# VCM: Vision Concept Modeling with Adaptive Vision Token Compression via Instruction Fine-Tuning

**Run Luo**[1,2,3]    **Renke Shan**[4]    **Longze Chen**[1,2]    **Ziqiang Liu**[1,2]    **Lu Wang**[4]
**Min Yang**[1,3*]    **Xiaobo Xia**[4,5*]

[1] Shenzhen Key Laboratory for High Performance Data Mining,
Shenzhen Institute of Advanced Technology, Chinese Academy of Sciences
[2]University of Chinese Academy of Sciences
[3]Shenzhen University of Advanced Technology
[4]National University of Singapore
[5]MoE Key Laboratory of Brain-inspired Intelligent Perception and Cognition,
University of Science and Technology of China
{R.LUO@SIAT.AC.CN    MIN.YANG@SIAT.AC.CN    XIAOBOXIA.UNI@GMAIL.COM}

## Abstract

Large vision-language models (LVLMs) have emerged as foundational tools for real-world AI applications. Despite their remarkable capabilities, current LVLMs process entire images at the token level, leading to significant inefficiencies compared to human cognition, which selectively focuses on high-level vision concepts. This token-level redundancy becomes increasingly problematic for high-resolution images and long video sequences, resulting in large computational costs and limited scalability in practical applications. To address this limitation, we introduce the concept of a vision concept model, a novel paradigm that enables LVLMs to dynamically extract the most relevant vision concepts from complex inputs, based on task-specific instructions. To optimize this vision concept modeling process, we propose VCM, a self-supervised framework that leverages vision-language correlations across diverse instances. VCM is designed to learn meaningful vision concepts without the need for expensive concept-level annotations. At its core, it employs a forward-backward optimization algorithm that supports LVLMs to adjust concept granularity and spatial alignment dynamically. Experiments demonstrate that VCM remarkably reduces computational costs (*e.g.*, achieving up to 85% fewer FLOPs for LLaVA-1.5-7B), while maintaining strong performance across a series of vision-language tasks. The codebase is available at https://github.com/RainBowLuoCS/VCM.

## 1   Introduction

Large vision-language models (LVLMs) [1, 2, 3, 4, 5, 6, 7, 8] play a critical role in addressing a wide range of vision-language tasks and have become a cornerstone for enabling general artificial intelligence to interact with the real world, such as in embodied intelligence [9, 10] and autonomous driving [11, 12, 13]. However, current LVLMs process entire images at the token level, which is inefficient compared to humans, who analyze information and generate content at the conceptual level, extracting relevant vision concepts with minimal effort. This inefficiency becomes particularly pronounced when dealing with higher-resolution images or longer video inputs, which humans can easily handle, but LVLMs struggle with it due to the rapidly increasing computational cost. This

---

[*]Corresponding authors.

39th Conference on Neural Information Processing Systems (NeurIPS 2025).

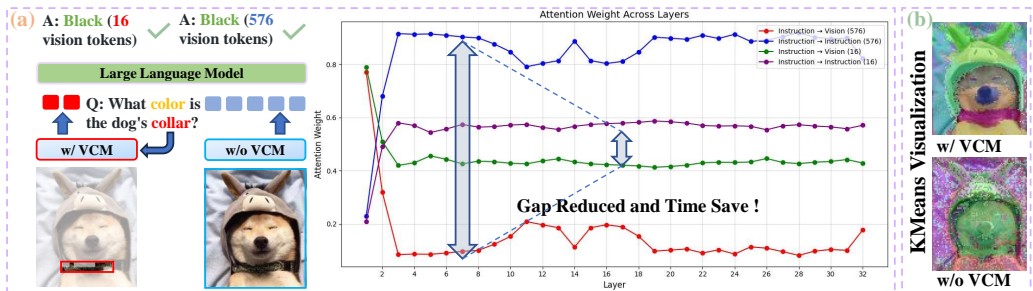

Figure 1: **Illustrations of VCM in enhancing efficiency and dense perception capability.** VCM can select relevant vision concept tokens based on instructions, significantly reducing redundant attention computations in LVLMs. This reduction lowers both training and inference costs while maintaining strong performance, as shown in (a). Additionally, VCM enhances the dense concept prediction capability of the vision encoder, as illustrated in (b) through K-Means visualizations of dense feature maps from the last layer of CLIP ViT. These improvements enable broader applicability to dense perception vision-language tasks.

limitation, arising from the lack of a vision concept model, restricts the usability of LVLMs in practical applications.

In this work, we formally define the concept of a **vision concept model**. The vision concept model can dynamically determine the required vision concepts, including their quantity and spatial locations, based on given instructions. Note that previous methods [14, 15, 16] have attempted to improve the efficiency of LVLMs by compressing vision tokens using variable-length queries or pruning the tokens through attention mechanisms. However, these methods fail to provide semantically meaningful concepts grounded in the corresponding spatial locations of vision inputs, making them unsuitable for vision-language tasks beyond visual question answering[2]. As such, they cannot be categorized as vision concept models.

To bridge this gap, we propose VCM, a self-supervised vision concept modeling framework. Specifically, VCM leverages vision-language correlations across multiple sampled instances and employs vision-language fine-tuning to build a vision concept model, without costly concept-level annotations. To enable the optimization with theoretical support for the vision tokens of varying lengths, we design a forward-backward algorithm based on the dynamic programming process [17], which supports adaptive and dynamic length optimization. This allows the vision concept model to dynamically output required vision concepts and keep their corresponding spatial locations according to given instructions, as shown in Figure 1. Benefiting from the vision concept model, VCM not only significantly reduces the computational cost of training and inference in LVLMs (*e.g.*, 85% fewer FLOPs for LLaVA-1.5-7B) while maintaining strong visual question answering performance but also enhances the vision encoder's capabilities in other dense perception vision-language tasks, such as zero-shot image classification, open-vocabulary object detection, and open-vocabulary semantic segmentation.

Before delving into details, our main contributions can be summarized as follows. (1) We formally define the vision concept model, which dynamically determines the required vision concepts, including their quantity and spatial locations, based on given instructions. This model is applicable to a variety of vision-language tasks. (2) We propose the VCM framework, which employs the correlations between vision and language across multiple sampled instances and a theoretically supported forward-backward algorithm to enable dynamic vision concept learning without requiring expensive fine-grained annotations. (3) We conduct extensive qualitative analyses, quantitative experiments, and ablation studies to validate the effectiveness and efficiency of VCM.

---

[2]We review related works and detail their difference from our method in Appendix A.

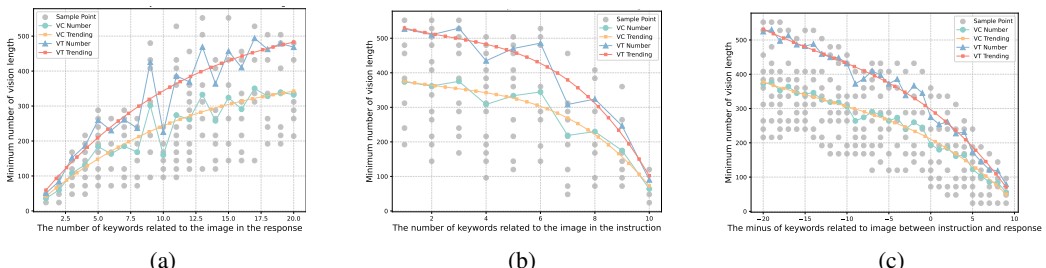

| (a) | (b) | (c) |

Figure 2: **Correlations between vision tokens and text prior.** (a) Positive correlation between the text response and minimum vision tokens: the more image-related keywords in the text response, the longer the minimum token length required. (b) Negative correlation between the text instruction and minimum vision tokens: the more image-related keywords in the text instruction, the shorter the minimum token length required. (c) Negative correlation between the difference in text response and instruction and minimum vision tokens: the greater the gap in image-related keywords between the text response and instruction, the shorter the minimum vision token length required.

## 2  Preliminaries

**Problem setting.** The LVLM architecture generally consists of a vision encoder $f_V$, a modality projector $f_M$, and a large language model (LLM) $f_{LLM}$ with its language model head (LMH) $f_{LMH}$. The vision encoder, typically a pre-trained image encoder like CLIP's vision model [18], converts input images $\mathbf{X}^V$ into vision tokens $f_V(\mathbf{X}^V)$. The projector aligns these vision tokens with text instruction $\mathbf{X}^I$ encoded by the LLM's word embedding $\mathbf{X}^T$, which enables the LLM to process vision data effectively. For simplicity, we define that $\mathbf{H}^V = f_M(f_V(\mathbf{X}^V)) \in \mathbb{R}^{M \times d}$ and $\mathbf{H}^I = \mathbf{X}^I(\mathbf{X}^T)^\top \in \mathbb{R}^{N \times d}$, where $M$ denotes the length of vision tokens, $N$ denotes length of text tokens , and $d$ is the hidden dimension size. That is usually $M \gg N$. The LLM then integrates the aligned vision information and instructions to produce response $\mathbf{X}^R$, which is learned as $p_\theta(\mathbf{X}^R|f_{LMH}(f_{LLM}([\mathbf{H}^V; \mathbf{H}^I])))$, parameterized by $\theta$. Note that due to a large number of vision tokens, the training and inference of LVLMs are inefficient [19]. Therefore, if a vision concept model exists that can decrease the value of $M$, the efficiency of LVLMs during both training and inference will be improved[3].

**Goal and challenges.** The primary goal of vision concept modeling (VCM) is to learn relevant vision conceptual information $\mathbf{H}_C^V$ from $\mathbf{H}^V$[4], based on language priors. The vision conceptual information can be applied to various vision-language tasks, leading to significant improvements in performance or efficiency. There are two key challenges in VCM. (1) Fine-grained labeled data is scarce, as manual annotation is labor-intensive, time-consuming, and inefficient. This makes it difficult to leverage the abundant unlabeled visual question answering (VQA) datasets effectively. (2) The length of vision concepts varies dynamically across different examples, making it challenging to model and optimize. Below we present how to handle the two challenges and implement VCM.

## 3  Vision Concept Modeling

### 3.1  Correlations between Vision Tokens and Text Prior

The proposed VCM utilizes the correlations between minimum required vision tokens and text priors. For instance, a general instruction like "*Can you describe this image in detail?*" typically demands information from the entire image, resulting in longer and keyword-rich responses. In contrast, a specific prompt like "*Where is the person in yellow?*" focuses on a distinct region, requiring less vision information and yielding shorter responses. To address the challenge of limited fine-grained annotations and to explore this relationship, we sample 5K instances from the LLaVA instruction fine-tuning dataset [20] and use GPT-4o to identify image-related keywords in both responses and instructions. We then applied VisionZip [16] with 24 different vision token lengths to generate

---

[3]We provide more analysis of the efficiency improvement from the floating-point operation (FLOP) perspective. Interested readers can check Appendix D for details.

[4]Here $\mathbf{H}_C^V$ is a sparse version of $\mathbf{H}^V$, with $\|\mathbf{H}_C^V\|_0 \ll \|\mathbf{H}^V\|_0$.

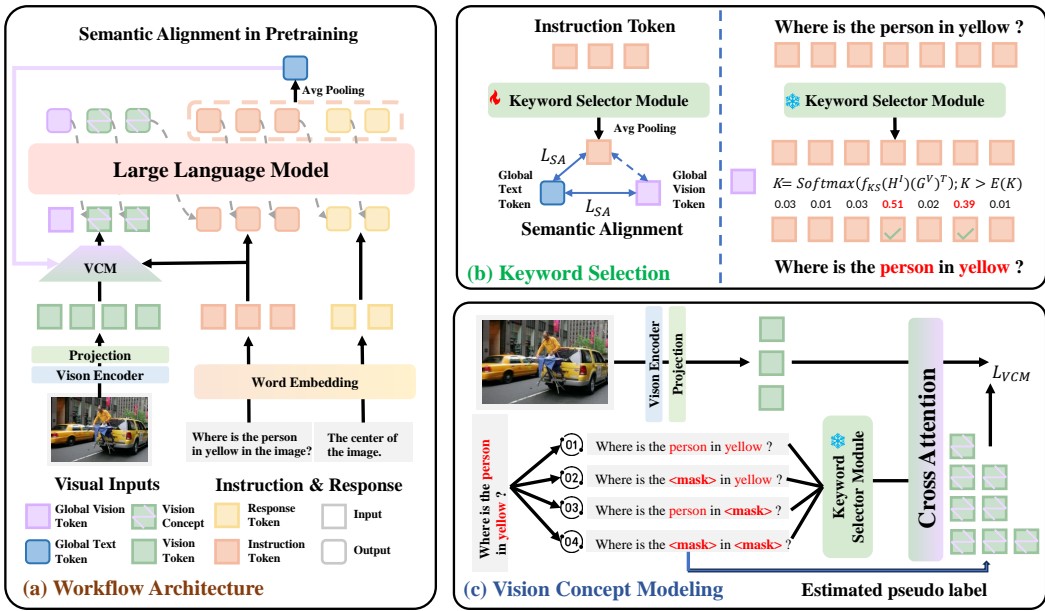

Figure 3: **Overview of our VCM framework.** (a) The workflow architecture: vision concepts are extracted from image inputs based on instruction priors and fed into the LLM to generate corresponding answers. (b) Adaptive keyword selection module: image-relevant keywords (highlighted in red) are selected by calculating text-image similarity, retaining keywords with scores above the average. (c) Implicit contrastive sampling module: keywords in the instruction are randomly masked, and the VCM loss is computed with input image features for end-to-end optimization.

responses via the LLaVA model, using GPT-4o as a judge to determine the minimum necessary token length. Experimental results in Figure 2 indicate a negative correlation between keyword count in instructions and the required vision token length, while the keyword count in responses showed a positive correlation. Furthermore, the difference in keyword counts between responses and instructions reduces noise and exhibits a more stable negative correlation. We also estimated the vision concept length by scaling the vision token length by $\sqrt{2}$ after local merging. This adjustment shows a more consistent correlation with text priors, which provides an efficient way to estimate vision concept length for self-supervised modeling.

### 3.2 Semantic Alignment for Keyword Selection

To efficiently and accurately extract keywords, we designed a keyword selector module $f_{\mathrm{KS}}$ composed of a multi-head self-attention (MHSA) layer for semantic alignment optimization. During the pre-training phase, we obtain the global text features $\mathbf{G}^{\mathrm{T}}$, the global vision features $\mathbf{G}^{\mathrm{V}}$, and the global language model text features $\mathbf{G}_{\mathrm{LLM}}^{\mathrm{T}}$ through keyword selector module $f_{\mathrm{KS}}$, vision encoder $f_{\mathrm{V}}$, and large language model $f_{\mathrm{LLM}}$ by global average pooling, respectively. Formally, we have $\mathbf{G}^{\mathrm{T}} = \mathrm{AvgPool}(\mathrm{MHSA}([\mathbf{X}^{\mathrm{I}};\mathbf{X}^{\mathrm{R}}](\mathbf{X}^{\mathrm{T}})^{\top}))$, $\mathbf{G}^{\mathrm{V}} = f_{\mathrm{V2T}}(\mathrm{AvgPool}(f_{\mathrm{V}}(\mathbf{X}^{\mathrm{V}})))$, and $\mathbf{G}_{\mathrm{LLM}}^{\mathrm{T}} = \mathrm{AvgPool}(f_{\mathrm{LLM}}([\mathbf{X}^{\mathrm{I}};\mathbf{X}^{\mathrm{R}}]))$, where $[;]$ denotes the concatenate operation, and $f_{\mathrm{V2T}}$ represents the vision to text linear project in modality project $f_{\mathrm{M}}$. Then, we map $\mathbf{G}^{\mathrm{T}}$, $\mathbf{G}^{\mathrm{V}}$ and $\mathbf{G}_{\mathrm{LLM}}^{\mathrm{T}}$ into semantic space by the language model head $f_{\mathrm{LMH}}$ and calculate the semantic alignment (SA) loss as

$$\mathcal{L}_{\mathrm{SA}} = -p(f_{\mathrm{LMH}}(\mathbf{G}_{\mathrm{LLM}}^{\mathrm{T}}))\log p(f_{\mathrm{LMH}}(\mathbf{G}^{\mathrm{V}})) - p(f_{\mathrm{LMH}}(\mathbf{G}_{\mathrm{LLM}}^{\mathrm{T}}))\log p(f_{\mathrm{LMH}}(\mathbf{G}^{\mathrm{T}})).$$

To better incorporate text prior, we adopt a multi-head cross-attention (MHCA) layer followed by a vision-to-text linear layer $f_{\mathrm{V2T}}$ as a modality projector $f_{\mathrm{M}}$ to extract vision information and align it to the text space as follows:

$$\mathbf{S} = \mathrm{AvgPool}(f_{\mathrm{KS}}(\mathbf{X}^{\mathrm{I}}(\mathbf{X}^{\mathrm{T}})^{\top})) - \mathrm{AvgPool}(f_{\mathrm{KS}}(\mathbf{X}^{\mathrm{R}}(\mathbf{X}^{\mathrm{T}})^{\top})), \mathbf{H}^{\mathrm{V}} = f_{\mathrm{V2T}}(\mathrm{MHCA}(\mathbf{Q}^{\mathrm{V}} \oplus f_{\mathrm{T2V}}(\mathbf{S}), f_{\mathrm{V}}(\mathbf{X}^{\mathrm{V}}))),$$
(1)

where $\mathbf{S}$ denotes the difference of instruction and response, $f_{\text{T2V}}$ represents the text-to-vision linear transformation used for learnable vision query $\mathbf{Q}^{\text{V}}$, and $\oplus$ means the broadcast add operation. Then the total loss in the pretraining stage can be defined as:

$$\mathcal{L}_{\text{total}}^{\text{I}} = -\log p(\mathbf{X}^{\text{R}}|f_{\text{LMH}}(f_{\text{LLM}}([\mathbf{G}^{\text{V}}; \mathbf{H}^{\text{V}}; \mathbf{H}^{\text{I}}]))) + \alpha \cdot \mathcal{L}_{\text{SA}},$$

where $\mathbf{H}^{\text{I}} = \mathbf{X}^{\text{I}}(\mathbf{X}^{\text{T}})^{\top}$ and $\alpha$ is the constant coefficient setting to 0.05. In the instruction fine-tuning phase, the frozen keyword selector can efficiently extract accurate keywords, which are then used for vision concept modeling. For example, the keywords in the instruction and response are adaptively determined using the following formula:

$$K = \texttt{Softmax}(\texttt{MHSA}([\mathbf{H}^{\text{I}}; \mathbf{H}^{\text{R}}])(\mathbf{G}^{\text{V}})^{\top}) = \texttt{Softmax}(\texttt{MHSA}([\mathbf{X}^{\text{I}}; \mathbf{X}^{\text{R}}](\mathbf{X}^{\text{T}})^{\top})(\mathbf{G}^{\text{V}})^{\top}), \quad (2)$$

where an indicator function $\mathbb{I}_K(k > \mathbb{E}(K))$ is used to extract the keyword set from $\mathbf{X}^{\text{I}}$ and $\mathbf{X}^{\text{R}}$. Figure 3(b) illustrates the process involved in the keyword selection component.

## 3.3 Vision Token Selection via Dynamic Programming

To tackle the challenge of dynamically varying lengths of vision tokens, we propose a forward-backward algorithm for dynamic vision token selection. Specifically, for aligned vision tokens $\mathbf{H}^{\text{V}}$ of length $M$, given text priors $\mathbf{T}$ with keyword number difference $N_{\text{key}}$ between instruction and response, we use a binary classification head $f_{\text{CLS}}$ to obtain classification results as follows:

$$\mathbf{Y}^{\text{V}} = f_{\text{CLS}}(\mathbf{H}^{\text{I}}) = f_{\text{CLS}}((f_{\text{V2T}}(\texttt{MHCA}(\mathbf{Q}^{\text{V}} \oplus f_{\text{T2V}}(\mathbf{T}), f_{\text{V}}(\mathbf{X}^{\text{V}}))))) = \{\mathbf{y}_t^{\text{V}}\}_{t=1}^{M}. \quad (3)$$

The length of the vision concept of the target is linearly estimated based on min-max normalization as $L = \lfloor M \cdot S \cdot (1 - \frac{N_{\text{key}} - N_{\text{key}}^{\min}}{N_{\text{key}}^{\max} - N_{\text{key}}^{\min}}) \rfloor$, where $N_{\text{key}}^{\max} = 10$, $N_{\text{key}}^{\min} = -35$, and $S$ is the information domain scalar used for maximum vision token length control. Additionally, when randomly masking keywords in the instructions with a mask ratio $r$ sampled from $\texttt{Uniform}(0, 1)$, the estimated vision concept length increases. This indicates that during the optimization process, the unsafe scenario of reducing necessary vision information and introducing additional hallucinations or noise does not occur. Therefore, such a random masking strategy not only enhances the learning of vision concept modeling but also provides a handle to control the vision token length in inference.

However, since we can only estimate the target length $L$ without knowing the exact positions of these features, there are multiple alignment possibilities. Exhaustively searching for all alignments has a time complexity of $\mathcal{O}(2^M)$, which is computationally prohibitive. To address this, we reformulate the problem into a dynamic programming process [17] with a time complexity of $\mathcal{O}(M^2)$ as below. Note that we have taken into this procedure may be somewhat complex mathematically. Therefore, we also provide an illustration for a better understanding (*cf.*, Table 6 of Appendix E).

**Extension of target sequence.** The target sequence $\mathbf{Z}^{\text{V}}$ is initialized as a sequence of $L$ symbols, where each symbol represents a retained vision concept $\mathbf{Z}^{\text{V}} = \{\mathbf{z}_i^{\text{V}}\}_{i=1}^{L}$ with $\mathbf{z}_i^{\text{V}} \in \{\star\}$, where $\star$ indicates a retained vision concept. The physical meaning of vision concepts here refers to a continuous segment of retained vision tokens. To account for possible alignments, the target sequence $\mathbf{Z}^{\text{V}}$ is extended by inserting blank symbols $\emptyset$ between and around the retained vision concept $\mathbf{Z}^{\text{V}} = [\emptyset, \mathbf{z}_1^{\text{V}}, \emptyset, \mathbf{z}_2^{\text{V}}, \emptyset, \ldots, \mathbf{z}_L^{\text{V}}, \emptyset]$. The length of the extended sequence $\mathbf{Z}^{\text{V}}$ is hence $2L + 1$.

**Forward probability initialization.** The forward variable $\alpha(t, l)$ represents the probability of aligning the first $t$ tokens of the input sequence $\mathbf{Y}^{\text{V}}$ to the first $l$ tokens of the extended target sequence $\mathbf{Z}^{\text{V}}$, where $\alpha(t, l) = p([\mathbf{z}_1^{\text{V}}, \cdots, \mathbf{z}_l^{\text{V}}]|[\mathbf{y}_1^{\text{V}}, \cdots, \mathbf{y}_t^{\text{V}}])$. The initialization is as follows. At the first time step $t = 1$, we set $\alpha(1, 1) = p(\emptyset|\mathbf{y}_1^{\text{V}})$ and $\alpha(1, 2) = p(\star|\mathbf{y}_1^{\text{V}})$, where $p(\emptyset|\mathbf{y}_1^{\text{V}})$ is the probability of aligning the first input token to the first blank symbol in the extended target sequence. For $l > 1$, the forward variable is initialized to $\alpha(1, l) = 0$ for $l > 2$.

**Forward probability transition.** The forward probabilities are computed iteratively from $t = 2$ to $M$ and $l = 1$ to $2L+1$ using the recurrence relation: $\alpha(t, l) = p(\mathbf{z}_l^{\text{V}}|\mathbf{y}_t^{\text{V}}) \cdot (\alpha(t-1, l) + \alpha(t-1, l-1))$, where $p(\mathbf{z}_l^{\text{V}}|\mathbf{y}_t^{\text{V}})$ is the probability of aligning the $t$-th input token to the $l$-th token in the extended target sequence, $p(\mathbf{z}_l^{\text{V}}|\mathbf{y}_t^{\text{V}}) \cdot \alpha(t-1, l)$ is the probability of staying at the current state, and $p(\mathbf{z}_l^{\text{V}}|\mathbf{y}_t^{\text{V}}) \cdot \alpha(t-1, l-1)$ is the probability of transitioning from the previous state.

**Backward probability initialization.** The backward variable $\beta(t, l)$ represents the probability of aligning the remaining tokens of the input sequence (from $t$ to $M$) to the remaining tokens of the extended target sequence (from $l$ to $2L + 1$), where $\beta(t, l) = p([\mathbf{z}_l^{\text{V}}, \cdots, \mathbf{z}_{2L+1}^{\text{V}}]|[\mathbf{y}_t^{\text{V}}, \cdots, \mathbf{y}_M^{\text{V}}])$. The initialization is as follows. At the last time step ($t = M$), we set $\beta(M, 2L + 1) = p(\emptyset|\mathbf{y}_M^{\text{V}})$ and $\beta(M, 2L) = p(\star|\mathbf{y}_M^{\text{V}})$, where $2L + 1$ is the final blank symbol in the extended target sequence. For $l < 2L + 1$, the backward variable is initialized to $\beta(M, l) = 0$ for $l < 2L$.

**Backward probability transition.** The backward probabilities are computed iteratively from $t = M - 1$ to 1 and $l = 2L$ to 1 using the following recurrence relation: $\beta(t, l) = (\beta(t + 1, l) + \beta(t + 1, l + 1)) \cdot p(\mathbf{z}_l^{\text{V}}|\mathbf{y}_t^{\text{V}})$, where $\beta(t + 1, l) \cdot p(\mathbf{z}_l^{\text{V}}|\mathbf{y}_t^{\text{V}})$ represents the probability of staying at the current state, and $\beta(t + 1, l + 1) \cdot p(\mathbf{z}_l^{\text{V}}|\mathbf{y}_t^{\text{V}})$ means the probability of transitioning to the next state.

### 3.4 Objective of VCM and Gradient Computation

The probability of the extended target sequence $\mathbf{Z}^{\text{V}}$ given the input sequence $\mathbf{Y}^{\text{V}}$ is computed as:

$$p(\mathbf{Z}^{\text{V}}|\mathbf{Y}^{\text{V}}) = \alpha(M, 2L) + \alpha(M, 2L + 1) = \beta(1, 1) + \beta(1, 2) = \sum_{l=1}^{2L+1} \alpha(t, l) \cdot \beta(t, l), \forall t \in [1, M].$$

(4)

The loss is then defined as the negative log-likelihood

$$\mathcal{L}_{\text{VCM}} = -\log p(\mathbf{Z}^{\text{V}}|\mathbf{Y}^{\text{V}}).$$

(5)

The gradient of the loss with respect to logits $\mathbf{y}_t^{\text{V}}(\mathbf{z}_l^{\text{V}})$ is computed using the posterior probability $\gamma(t, l)$, $p(\mathbf{z}_l^{\text{V}}|\mathbf{y}_t^{\text{V}})$ represents the probability of aligning the $t$-th input token to the $l$-th token in the extended target sequence $\gamma(t, l) = \alpha(t, l) \cdot \beta(t, l)/p(\mathbf{Z}^{\text{V}}|\mathbf{Y}^{\text{V}})$. The gradient is then given by: $\frac{\partial \mathcal{L}_{\text{VCM}}}{\partial \mathbf{y}_t^{\text{V}}(\mathbf{z}_l^{\text{V}})} = p(\mathbf{z}_l^{\text{V}}|\mathbf{y}_t^{\text{V}}) - \gamma(t, l)$ (see detailed mathematical derivations in Appendix C), where an indicator function $\mathbb{I}_{\mathbf{Y}^{\text{V}}}(p(\star|\mathbf{y}_t^{\text{V}}) > p(\emptyset|\mathbf{y}_t^{\text{V}}))$ is used to extract the vision concept $\mathbf{H}_{\text{C}}^{\text{V}}$ from $\mathbf{H}^{\text{V}}$ via a segment merging (SM) operation $\mathbf{H}_{\text{C}}^{\text{V}} = \text{SM}(\mathbf{H}^{\text{V}}, \mathbf{Y}^{\text{V}}, \mathbb{I}_{\mathbf{Y}^{\text{V}}}(p(\star|\mathbf{y}_t^{\text{V}}) > p(\emptyset|\mathbf{y}_t^{\text{V}}))$.

For better understanding, we provide detailed pseudocodes for the training pipeline of $\mathcal{L}_{\text{VCM}}$ in Algorithm 1 and for the SM operation in Algorithm 2 respectively (check Appendix E for more details). The weighted average method is used to extract the concept level features $\mathbf{H}_{\text{C}}^{\text{V}}$ from the features $\mathbf{H}^{\text{V}}$ of the token level, giving the physical meaning of the importance of the score $\mathbf{Y}^{\text{V}}$, which is more suitable for other scenarios. The next token prediction loss is defined as:

$$\mathcal{L}_{\text{NTP}} = -\log p(\mathbf{X}^{\text{R}}|f_{\text{LMH}}(f_{\text{LLM}}([\mathbf{G}^{\text{V}}; \mathbf{H}_{\text{C}}^{\text{V}}; \mathbf{H}^{\text{I}}]))).$$

The total loss in the instruction fine-tuning stage is then defined as

$$\mathcal{L}_{\text{total}}^{\text{II}} = \mathcal{L}_{\text{NTP}} + \epsilon(r) \cdot \mathcal{L}_{\text{VCM}},$$

where $\epsilon(r)$ is a coefficient function used to balance two loss functions. Note that when the mask ratio $r$ is small, the VCM loss is not so accurate, the coefficient is appropriately small. Besides, when $r$ is large, the VCM loss is more accurate and appropriately increases, using the hyperbolic tangent smoothing function. More mathematical details about $\epsilon(r)$ can be found in Appendix B. After vision concept modeling, we can effectively control the model to precisely output vision concepts of different lengths based on the mask ratio $r$.

## 4 Experiments

### 4.1 Experimental Setups

**Datasets.** To justify our claims and demonstrate the superiority of VCM, several representative tasks are involved, which include visual question answering, zero-shot image classification, open-vocabulary object detection, open-vocabulary semantic segmentation, and video understanding. Specifically, for visual question answering (VQA) evaluation, we conduct experiments on 11 widely adopted image-based benchmarks, including VQAV2 (VQA$^{\text{V2}}$) [21], GQA [22], VisWiz [23],

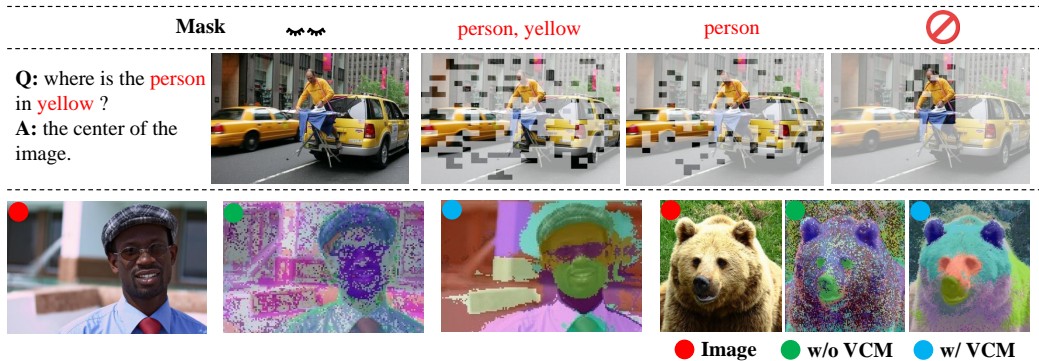

Figure 4: **Visualization results about our VCM**. (*Top*) Visualization of VCM with different instructions. From left to right, the visual representation becomes increasingly sparse, leaving corresponding vision tokens to unmasked keywords (highlighted in red). (*Bottom*) K-Means visualization of dense feature maps of CLIP ViT. We show the raw images, the K-Means results without VCM, and those of our fine-tuned model by VCM.

SciQA [24], POPE [25], MME [26], MMBench (MMB) [27], SEED-Image (SEED) [28], MMvet [29], TextVQA (VQA$^T$) [30], and MMStar [31]. Also, RefCOCO [32] is used for the region-level VQA task. Besides, we employ COCO [33] and OV-COCO [34] for zero-shot image classification and open-vocabulary object detection, and use ADE-150 [35] and ADE-847 [35] for open-vocabulary semantic segmentation. To test the video understanding capability of VCM, 4 common video question answering benchmarks, TGIF-QA [36], MSVD-QA [37], MSRVTT-QA [37], and ActivityNet-QA [38], are included. Here LMMs-Eval toolkit [39] is employed for fair comparison. More details of the tasks and datasets are included in Appendix E.

**Models.** We follow the architecture from the LLaVA series [20, 1], where an LVLM consists of three key components: an LLM for next token prediction, a vision encoder for extracting vision features, and an image-text projector for vision and language alignment. Besides, we utilize a multi-head self-attention layer for the keyword selector module and a multi-head cross-attention layer for vision concept modeling, both employing 16 heads. For LLaVA-1.5-7/13B, LLaVA-NeXT [1], and Video-LLaVA [40], we adopt the original paper's publicly available inference setting.

**Training strategies.** We follow the widely used two-stage setting for training. Specifically, it includes vision-language pre-training and instruction fine-tuning. The language models and ViT are separately pre-trained, while the projector is randomly initialized. To initially align the feature space between vision and language modalities, we utilize an aligned dataset [20]. Afterward, we perform instruction fine-tuning of the pre-trained model on vision-language instruction datasets. The AdamW optimizer [41] is exploited, with learning rates $5 \times 10^{-5}$ and $2 \times 10^{-5}$ for aforementioned two stages respectively. The instruction fine-tuning stage is trained with two epochs with a 3% warmup strategy. The experiments are conducted with 8×NVIDIA A100-80G GPUs.

### 4.2 Qualitative Analysis

As shown in Figure 4(*top*), we visualize the performance of VCM on visual question answering (VQA) examples. From left to right, we demonstrate the results of masking different numbers of keywords in the instruction. As fewer keywords (highlighted in red) are masked, more irrelevant tokens are removed, and the vision concepts are progressively refined. The visualizations indicate that, although VCM discards some overall image details, it effectively retains key vision concepts required for correctly answering the question. More visualization cases can be found in Appendix G.

We further demonstrate the vision concept representation of the vision encoder trained with VCM by performing K-Means clustering [42] on dense feature maps. Specifically, pixels with high cosine similarity are grouped into the same cluster. For clarity, clusters containing only a small

Table 1: Performance on 11 image-based VQA benchmarks. '#Vision Tokens' is the number of vision tokens fed to the LLM backbone. '*' indicates that unfreezing the image encoder in the instruction fine-tuning stage.

| Methods | #Vision Tokens | $\text{VQA}^{v2}$ | GQA | VisWiz | SciQA | POPE | MME | MMB | SEED | MM-Vet | $\text{VQA}^T$ | MMstar | Avg. (%) |
|---|---|---|---|---|---|---|---|---|---|---|---|---|---|
| BLIP-2 [43] | 32 | 65.0 | 41.0 | 19.6 | 61.0 | 85.3 | 1293.8 | 31.2 | 46.4 | 22.4 | – | – | – |
| InstructBLIP [3] | 32 | 72.4 | 49.2 | 34.5 | 60.5 | 86.1 | 1391.4 | 36.0 | 53.4 | 26.2 | 33.6 | 32.7 | 48.6 |
| IDEFICS-9B [44] | 64 | 50.9 | 38.4 | 35.5 | 53.5 | 81.9 | 1177.3 | 48.2 | 45.0 | 30.0 | – | 21.6 | – |
| IDEFICS-80B [44] | 64 | 60.0 | 45.2 | 36.0 | 61.8 | 66.0 | 1518.2 | 54.5 | 52.1 | 39.7 | – | 26.1 | – |
| Qwen-VL [2] | 256 | 78.8 | 59.3 | 35.2 | 67.1 | 70.0 | 482.7 | 38.2 | 56.3 | 13.0 | 63.1 | 32.5 | 48.2 |
| Qwen-VL-Chat [2] | 256 | 78.2 | 57.5 | 38.9 | 68.2 | 74.9 | 1860.0 | 60.6 | 58.2 | 47.3 | 60.7 | 34.5 | 58.7 |
| SPHINX [45] | 289 | 78.1 | 62.6 | 39.9 | 69.3 | 80.7 | 1826.1 | 66.9 | 56.2 | 36.0 | – | – | – |
| mPLUG-Owl2 [46] | 1024 | 79.4 | 56.1 | 54.5 | 68.7 | 84.6 | 1786.4 | 64.5 | 57.8 | 36.2 | 56.4 | 34.8 | – |
| LLaVA-v1.5 [20] | 576 | 78.5 | 62.0 | 50.0 | 66.8 | 85.9 | 1862.1 | 64.3 | 58.6 | 30.5 | 58.2 | 33.2 | 59.5 |
| LLaVA-v1.5* [20] | 576 | 77.6 | 61.7 | 46.5 | 66.5 | 86.0 | 1793.3 | 64.1 | 57.3 | 30.2 | 57.8 | 33.0 | 58.6 |
| *LVLMs with vision token reduction methods* | | | | | | | | | | | | | |
| PruMerge [47] | 32 | 72.0 | – | – | 68.5 | 76.3 | 1350.3 | 60.9 | – | – | – | – | – |
| PruMerge++ [47] | 144 | 76.8 | – | – | 68.3 | 84.0 | 1462.4 | 64.9 | – | – | – | – | – |
| MQT-LLaVA [14] | 144 | 76.4 | 61.4 | 52.0 | 67.5 | 83.9 | 1798.2 | 64.4 | 60.2 | 29.2 | 52.4 | 32.6 | 58.6 |
| FastV [19] | 192 | 67.1 | 52.7 | 46.7 | 65.2 | 64.8 | 1612.0 | 61.2 | 57.1 | 27.7 | 52.5 | 31.7 | 53.1 |
| SparseVLM [15] | 192 | 75.6 | 57.6 | 51.6 | 69.1 | 83.6 | 1721.0 | 62.5 | 55.8 | 31.5 | 56.1 | 32.0 | 57.9 |
| PDrop [48] | 192 | 76.1 | 57.1 | 51.3 | 70.2 | 82.3 | 1766.0 | 63.2 | 54.7 | 30.5 | 56.1 | 32.1 | 57.8 |
| VisionZip [16] | 192 | 77.4 | 60.1 | 51.2 | 68.2 | 84.9 | 1834.0 | 63.4 | 57.1 | 32.6 | 57.8 | 32.3 | 59.1 |
| **Ours** | | | | | | | | | | | | | |
| VCM | 144 | 77.2 | 61.9 | 54.9 | 69.1 | 86.1 | 1842.7 | 64.5 | 64.3 | 32.4 | 57.4 | 35.7 | 60.8 |
| Δ *vs. LLaVA-v1.5* | | -1.3 | -0.1 | +4.9 | +2.3 | +0.2 | -19.4 | +0.2 | +5.7 | +1.9 | -0.8 | +2.5 | +1.3 |
| VCM* | 144 | 76.8 | 61.4 | 54.3 | 68.7 | 85.9 | 1823.6 | 64.2 | 62.7 | 31.6 | 57.2 | 35.4 | 60.3 |
| Δ *vs. LLaVA-v1.5** | | -0.8 | -0.3 | +7.8 | +2.2 | -0.1 | +30.3 | +0.1 | +5.4 | +1.4 | -0.6 | +2.4 | +1.7 |

Table 2: Results of region-level VQA and zero-shot image classification tasks. The best result in each case is shown in bold.

| Methods | CLIP | VCM | Region-level VQA (RefCOCO) | | | | | | Image Classification (COCO) | | | |
|---|---|---|---|---|---|---|---|---|---|---|---|---|
| | | | BBox | | | Mask | | | Thing Masks | | Stuff Masks | |
| | | | testA | testB | val | testA | testB | val | Top 1 | Top 5 | Top 1 | Top 5 |
| LLaVA-v1.5 | ViT-B/16 | × | 16.6 | 43.7 | 33.2 | 16.8 | 43.5 | 33.6 | 33.5 | 56.0 | 25.9 | 50.9 |
| LLaVA-v1.5 | ViT-B/16 | √ | **18.9** | **45.4** | **35.7** | **19.1** | **45.6** | **35.4** | **42.7** | **70.6** | **39.8** | **64.6** |
| LLaVA-v1.5 | ViT-L/14 | × | 14.9 | 40.1 | 29.5 | 14.4 | 40.3 | 29.6 | 28.3 | 52.0 | 11.8 | 27.9 |
| LLaVA-v1.5 | ViT-L/14 | √ | **16.5** | **41.3** | **31.8** | **16.8** | **41.6** | **31.9** | **43.8** | **69.7** | **25.4** | **47.0** |

number of pixels are removed. As shown in Figure 4(*bottom*), the CLIP ViT trained with VCM exhibits significantly improved performance in grouping pixels belonging to the same object into a single cluster, compared to the baseline without VCM. For example, in the visualization, clusters corresponding to "face", "hat", and "glasses" are notably more accurate. These K-Means clustering results provide intuitive evidence for the enhanced vision concept representation capability of CLIP ViT. Additional visualization cases can also be checked in Appendix G.

## 4.3 Benchmark Comparison

To evaluate the effectiveness of VCM, we conduct experiments on 11 widely used image-based VQA benchmarks and compare our method against existing token reduction state-of-the-art (SOTA) methods [19, 15, 16, 14, 48]. Then, we explore VCM in enhancing the fine-grained perception capabilities of LVLMs on and region-level VQA and zero-shot image classification tasks. Finally, we validate the broad applicability of VCM in improving other vision-language tasks such as open-vocabulary object detection and open-vocabulary semantic segmentation. We provide additional experiments about the acceleration effects of VCM in high-resolution scenarios and video understanding tasks, as well as its generalization and scalability across different architectures in Appendix F.

**Results of image-based VQA.** As shown in Table 1, we apply VCM to train LLaVA-1.5 and then evaluate achieved performance. When retaining only 144 tokens, our method outperforms PDrop [48], SparseVLM [15], and VisionZip [16] with fewer vision length by 5.2%, 5.0%, and 2.9%, respectively. Furthermore, VCM outperforms the vanilla model with 144 vision tokens retained. This means that our VCM method brings strong vision representation capabilities.

**Results of region-level VQA and zero-shot classification.** We study the performance of the LLaVA model trained with VCM on the region-level VQA task. Specifically, given a bounding box or mask

Table 3: Results of open-vocabulary (OV) object detection and open-vocabulary (OV) semantic segmentation tasks. The best result in each case is shown in bold.

| | | OV Object Detection | | | | | OV Semantic Segmentation | | | | |
| | | | OV-COCO | | | | | ADE-150 | | ADE-847 | |
| Method | CLIP | $AP_{50}^{novel}$ | $AP_{50}^{base}$ | $AP_{50}$ | Method | CLIP | mIoU | mAcc | mIoU | mAcc |
|---|---|---|---|---|---|---|---|---|---|---|
| – | – | – | – | – | SAN [51] | ViT-B/16 | 27.5 | 45.6 | 10.1 | 21.1 |
| – | – | – | – | – | SAN [51] | ViT-L/14 | 32.1 | 50.7 | 12.4 | 25.2 |
| F-VLM [52] | ViT-B/16 | 16.0 | 36.9 | 31.4 | Cat-Seg [53] | ViT-B/16 | 27.2 | 41.2 | 8.4 | 16.6 |
| F-VLM [52] | ViT-L/14 | 9.2 | 44.3 | 35.2 | Cat-Seg [53] | ViT-L/14 | 31.5 | 46.2 | 10.8 | 20.5 |
| F-VLM+VCM | ViT-B/16 | **20.6** | **47.1** | **38.6** | Cat-Seg+VCM | ViT-B/16 | **29.4** | **45.5** | **9.1** | **21.3** |
| F-VLM+VCM | ViT-L/14 | **12.6** | **48.7** | **39.2** | Cat-Seg+VCM | ViT-L/14 | **34.8** | **53.1** | **11.9** | **23.1** |

Table 4: Ablation study of the coefficient function $\epsilon(r)$, weighted average method in SM operation, semantic alignment, mask strategy, and information domain scalar $S$ on 4 image-based VQA benchmarks. The best result in each case is shown in bold. The default setting used in the main experiments is marked as underline.

| $\epsilon(r)$ | weighted average | semantic alignment | mask strategy | $S$ | #max tokens | SciQA | VizWiz | POPE | MME | Avg. |
|---|---|---|---|---|---|---|---|---|---|---|
| × | × | × | × | 1/4 | 144 | 60.7 | 46.5 | 73.4 | 1231.5 | 56.1 |
| ✓ | × | × | × | 1/4 | 144 | 61.7 | 46.7 | 75.3 | 1235.2 | 57.0 |
| ✓ | ✓ | × | × | 1/4 | 144 | 62.4 | 47.6 | 75.8 | 1334.7 | 58.4 |
| ✓ | ✓ | ✓ | × | 1/4 | 144 | 63.1 | 47.8 | 76.2 | 1344.2 | 58.7 |
| ✓ | ✓ | ✓ | ✓ | 1/2 | 288 | 62.9 | 46.9 | **76.8** | **1369.7** | 58.9 |
| ✓ | ✓ | ✓ | ✓ | 1/4 | 144 | **64.1** | **48.2** | 76.6 | 1350.6 | **59.3** |
| ✓ | ✓ | ✓ | ✓ | 1/6 | 72 | 61.1 | 46.4 | 74.0 | 1234.9 | 56.4 |
| ✓ | ✓ | ✓ | ✓ | 1/8 | 36 | 61.7 | 46.6 | 76.1 | 1244.1 | 57.2 |

label, we highlight the corresponding region in the image and send the image to the model for VQA related to that region. As shown in Table 2, using VCM significantly enhances the ability of LVLMs to understand vision concepts in specific regions of the image. These results are consistent with the phenomena observed in our qualitative analysis. Afterward, for zero-shot image classification on the COCO validation set [33], we extract dense feature maps from the CLIP ViT using panoptic masks (thing and stuff) from the COCO Panoptic dataset [49] and mask pooling operations [50]. The corresponding accuracy is reported in Table 2. The results show that VCM significantly improves the classification capabilities of CLIP ViT on panoptic masks.

**Results of open-vocabulary object detection and semantic segmentation.** We follow the setup of F-VLM [52], which freezes the CLIP-ViT backbone and extracts multi-scale feature maps for object detection. As shown in Table 3, replacing the CLIP ViT with the VCM-trained version significantly improves performance on OV-COCO [34] (*e.g.*, 9.2 vs. 12.6 for $AP_{50}^{novel}$, 44.3 vs. 48.7 for $AP_{50}^{base}$, and 35.2 vs. 39.2 for $AP_{50}$). These results highlight the effectiveness of VCM in open-vocabulary object detection. Besides, we apply the VCM fine-tuned model to CatSeg [54], where the dense features of CLIP ViT are utilized in a cost-aggregation module. The segmentation model is trained on the COCO Stuff dataset [53] and evaluated on ADE-150 and ADE-847. As shown in Table 3, VCM consistently improves performance across all test datasets. These results indicate that using VCM not only effectively accelerates the inference of LVLMs but also enhances the dense perception capabilities of the vision encoder.

## 4.4 Ablation Study

To investigate the impact of different components in our model design, we conduct a series of ablation studies on 4 image-based VQA benchmarks. Note that all experiments are performed with a batch size of 128. The number of training steps is set to 500 for fast evaluation.

As shown in Table 4, using the coefficient function can enhance overall model performance, which validates the effectiveness of this design. Then, incorporating a weighted averaging method in the concept extraction process, which integrates importance scores into next-token prediction optimiza-

tion, further improves overall model performance. Additionally, using semantic alignment training during the pre-training phase can improve overall performance, validating the effectiveness of the keyword selector module design. Furthermore, the introduction of a random masking strategy can serve as a data augmentation method, enhancing the effectiveness of VCM.

We also study the influence of the information domain. As demonstrated in Table 4, a larger information domain, which retains more visual information, generally leads to better performance. However, as the information domain size decreases, the shorter sampling length stabilizes training and can offer additional performance gains. Selecting an information domain size of 1/4 achieves a favorable trade-off between computational cost and performance, since it maintains competitive accuracy while reducing FLOPs. Based on these findings, we adopt the 1/4 information domain configuration across all experiments.

# 5   Conclusion

In this paper, we propose vision concept modeling (VCM) that efficiently extracts vision concepts for various vision-language tasks. VCM utilizes a large-scale vision-language instruction data with coarse supervision, employs implicit contrastive sampling, and integrates a dynamic concept extraction algorithm optimized through a forward-backward framework. Extensive experiments demonstrate that our method reduces computational costs while maintaining strong performance in a series of scenarios. Ablation studies further validate its effectiveness and scalability.

# Acknowledgements

Min Yang is supported by Guangdong Basic and Applied Basic Research Foundation (2025B1515020032 and 2024A1515030166). Xiaobo Xia is supported by MoE Key Laboratory of Brain-inspired Intelligent Perception and Cognition, University of Science and Technology of China (Grant No. 2421002).

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

# A  Related Work

## A.1  Large Vision-Language Models (LVLMs)

Benefiting from the success of large language models (LLMs) and the availability of diverse image-text instruction data [20, 55, 56, 57], LVLMs have made significant strides in recent years [58, 59, 60, 61, 62, 63, 64, 65]. For example, LLaVA [20] and MiniGPT-4 [66] have demonstrated strong cross-task generalization by integrating vision encoders with LLMs through simple connectors and training on instruction data. LLaVA-NeXT [1] has significantly enhanced visual perception by employing dynamic resolution techniques. DreamLLM [67] attempts to generate images and text in the interleaved context concurrently. DEEM [4] simplifies model architecture and enhances robustness by using diffusion models to extract vision features instead of traditional vision encoders. Readers can refer to [68, 69, 70, 71] for more details and recent advances in LVLMs.

## A.2  Vision Token Reduction for LVLMs

Vision token reduction is a key technique for improving the efficiency of LVLMs. Previous works, *e.g.*, EViT [72] and ToMe [73], lower computational costs by pruning less important tokens [74, 75, 76]. Differently, methods such as LLaVolta [77], Qwen-VL [2], and MQT-LLaVA [78], use clustering or Q-former to compress tokens into fixed-length representations, but these token merging methods fail to retain the relative order and positional relationships of the original tokens. Recent strategies like FastV [19], SparseVLM [15], PyramidDrop [48], and VisionZip [16] leverage language model signals and fixed pruning ratios to enhance efficiency. However, it is challenging for them to control the number of pruned vision tokens. Moreover, these methods often disrupt vision concept modeling due to over-compression, resulting in noise and misalignment between vision and semantic concepts, which restricts the model's applicability.

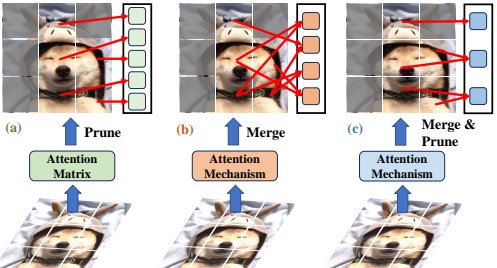

Figure 5: **Comparison of three different vision token reduction paradigms**. (a) Regular training-free token pruning method based on threshold filtering of the attention matrix. (b) Regular token merging method based on the attention mechanism and fixed-length trainable query vectors. (c) Our method, through adaptive local merging and filtering modeled by VCM, allows precise control of the vision concept length while keeping the positional information.

To overcome these concerns, we propose VCM, which uses rigorous theory-driven optimization to enable dynamic vision token selection and vision concept modeling. Our VCM not only allows precise control of the vision concept length, but also preserves the relative order and positional relationships of original vision tokens. As illustrated in Figure 5, compared to vision token compression paradigms (a) and (b), our method (c) achieves better results with a shorter and more controllable vision concept length, without losing position information or vision concept. VCM therefore can improve the efficiency of LVLMs and broaden their applicability to various vision-language tasks.

# B  Mathematical Expression of $\epsilon(r)$

The coefficient function $\epsilon(r)$ can be expressed mathematically as follows.

**Step 1.** Compute a smooth S-shaped function using the hyperbolic tangent ($\tanh$) function:

$$s = \frac{1 + \tanh\left(k \cdot (2 \cdot r - 1)\right)}{2}, \tag{6}$$

where $k$ is a parameter that controls the growth rate. Larger values of $k$ result in steeper transitions, while smaller values result in smoother transitions.

**Step 2.** Scale and shift $s$ to the desired range $[a, b]$:

$$\epsilon(r) = a + (b - a) \cdot s, \tag{7}$$

where $a = 0.2$, $b = 1.2$, and $k = 5$ in our default setting.

## C   Proof of Gradient Computation about $\mathcal{L}_{\text{VCM}}$

*Proof.* We provide the proof of the gradient of $\mathcal{L}_{\text{VCM}}$ with respect to $\mathbf{y}_t^{\text{V}}(\mathbf{z}_l^{\text{V}})$ step by step.

①**(Chain rule).** By definition,

$$\mathcal{L}_{\text{VCM}} = -\log p(\mathbf{Z}^{\text{V}} | \mathbf{Y}^{\text{V}}),$$

so

$$\frac{\partial \mathcal{L}_{\text{VCM}}}{\partial \mathbf{y}_t^{\text{V}}(\mathbf{z}_l^{\text{V}})} = -\frac{1}{p(\mathbf{Z}^{\text{V}} | \mathbf{Y}^{\text{V}})} \frac{\partial p(\mathbf{Z}^{\text{V}} | \mathbf{Y}^{\text{V}})}{\partial \mathbf{y}_t^{\text{V}}(\mathbf{z}_l^{\text{V}})}.$$

②**(Marginal probability and alignment decomposition).** By the forward-backward algorithm,

$$p(\mathbf{Z}^{\text{V}} | \mathbf{Y}^{\text{V}}) = \sum_{l'=1}^{2L+1} \alpha(t, l') \beta(t, l').$$

Only the terms with $l' = l$ depend on $\mathbf{y}_t^{\text{V}}(\mathbf{z}_l^{\text{V}})$, thus

$$\frac{\partial p(\mathbf{Z}^{\text{V}} | \mathbf{Y}^{\text{V}})}{\partial \mathbf{y}_t^{\text{V}}(\mathbf{z}_l^{\text{V}})} = \frac{\partial \alpha(t, l)}{\partial \mathbf{y}_t^{\text{V}}(\mathbf{z}_l^{\text{V}})} \beta(t, l) + \alpha(t, l) \frac{\partial \beta(t, l)}{\partial \mathbf{y}_t^{\text{V}}(\mathbf{z}_l^{\text{V}})}.$$

③**(Forward/backward variable dependence).** From the recursive definitions, both $\alpha(t, l)$ and $\beta(t, l)$ are linear in $p(\mathbf{z}_l^{\text{V}} | \mathbf{y}_t^{\text{V}})$, which is the softmax over logits $\mathbf{y}_t^{\text{V}}$. Thus, for fixed $(t, l)$,

$$\frac{\partial p(\mathbf{Z}^{\text{V}} | \mathbf{Y}^{\text{V}})}{\partial p(\mathbf{z}_l^{\text{V}} | \mathbf{y}_t^{\text{V}})} = \alpha(t, l) \beta(t, l) / p(\mathbf{z}_l^{\text{V}} | \mathbf{y}_t^{\text{V}}).$$

④**(Softmax chain rule).** By the softmax derivative,

$$\frac{\partial p(\mathbf{z}_k^{\text{V}} | \mathbf{y}_t^{\text{V}})}{\partial \mathbf{y}_t^{\text{V}}(\mathbf{z}_l^{\text{V}})} = p(\mathbf{z}_k^{\text{V}} | \mathbf{y}_t^{\text{V}})(\delta_{kl} - p(\mathbf{z}_l^{\text{V}} | \mathbf{y}_t^{\text{V}})).$$

where

$$\delta_{kl} = \begin{cases} 1, & \text{if } k = l \\ 0, & \text{if } k \neq l \end{cases}$$

⑤**(Final assembly, detailed).**

By the chain rule, the gradient of the loss with respect to the logits is

$$\frac{\partial \mathcal{L}_{\text{VCM}}}{\partial \mathbf{y}_t^{\text{V}}(\mathbf{z}_l^{\text{V}})} = \sum_{k=1}^{2L+1} \frac{\partial \mathcal{L}_{\text{VCM}}}{\partial p(\mathbf{z}_k^{\text{V}} | \mathbf{y}_t^{\text{V}})} \cdot \frac{\partial p(\mathbf{z}_k^{\text{V}} | \mathbf{y}_t^{\text{V}})}{\partial \mathbf{y}_t^{\text{V}}(\mathbf{z}_l^{\text{V}})}$$

where

$$\frac{\partial \mathcal{L}_{\text{VCM}}}{\partial p(\mathbf{z}_k^{\text{V}} | \mathbf{y}_t^{\text{V}})} = -\frac{1}{p(\mathbf{Z}^{\text{V}} | \mathbf{Y}^{\text{V}})} \cdot \frac{\partial p(\mathbf{Z}^{\text{V}} | \mathbf{Y}^{\text{V}})}{\partial p(\mathbf{z}_k^{\text{V}} | \mathbf{y}_t^{\text{V}})} = -\frac{\alpha(t, k) \beta(t, k)}{p(\mathbf{Z}^{\text{V}} | \mathbf{Y}^{\text{V}}) \, p(\mathbf{z}_k^{\text{V}} | \mathbf{y}_t^{\text{V}})}$$

Substituting the softmax derivative, we have

$$\frac{\partial \mathcal{L}_{\text{VCM}}}{\partial \mathbf{y}_t^{\text{V}}(\mathbf{z}_l^{\text{V}})} = \sum_{k=1}^{2L+1} \left[ -\frac{\alpha(t, k) \beta(t, k)}{p(\mathbf{Z}^{\text{V}} | \mathbf{Y}^{\text{V}}) \, p(\mathbf{z}_k^{\text{V}} | \mathbf{y}_t^{\text{V}})} \cdot p(\mathbf{z}_k^{\text{V}} | \mathbf{y}_t^{\text{V}})(\delta_{kl} - p(\mathbf{z}_l^{\text{V}} | \mathbf{y}_t^{\text{V}})) \right]$$

$$= -\sum_{k=1}^{2L+1} \frac{\alpha(t, k) \beta(t, k)}{p(\mathbf{Z}^{\text{V}} | \mathbf{Y}^{\text{V}})} (\delta_{kl} - p(\mathbf{z}_l^{\text{V}} | \mathbf{y}_t^{\text{V}}))$$

Now separate the $k = l$ term and the $k \neq l$ terms:

$$= -\frac{\alpha(t,l)\beta(t,l)}{p(\mathbf{Z}^{\mathrm{V}}|\mathbf{Y}^{\mathrm{V}})}(1 - p(\mathbf{z}_l^{\mathrm{V}}|\mathbf{y}_t^{\mathrm{V}})) + \sum_{k \neq l}\frac{\alpha(t,k)\beta(t,k)}{p(\mathbf{Z}^{\mathrm{V}}|\mathbf{Y}^{\mathrm{V}})}p(\mathbf{z}_l^{\mathrm{V}}|\mathbf{y}_t^{\mathrm{V}})$$

$$= -\frac{\alpha(t,l)\beta(t,l)}{p(\mathbf{Z}^{\mathrm{V}}|\mathbf{Y}^{\mathrm{V}})} + \left(\sum_{k=1}^{2L+1}\frac{\alpha(t,k)\beta(t,k)}{p(\mathbf{Z}^{\mathrm{V}}|\mathbf{Y}^{\mathrm{V}})}\right)p(\mathbf{z}_l^{\mathrm{V}}|\mathbf{y}_t^{\mathrm{V}})$$

Since $\sum_{k=1}^{2L+1}\frac{\alpha(t,k)\beta(t,k)}{p(\mathbf{Z}^{\mathrm{V}}|\mathbf{Y}^{\mathrm{V}})} = 1$, this simplifies to

$$\frac{\partial \mathcal{L}_{\mathrm{VCM}}}{\partial \mathbf{y}_t^{\mathrm{V}}(\mathbf{z}_l^{\mathrm{V}})} = p(\mathbf{z}_l^{\mathrm{V}}|\mathbf{y}_t^{\mathrm{V}}) - \frac{\alpha(t,l)\beta(t,l)}{p(\mathbf{Z}^{\mathrm{V}}|\mathbf{Y}^{\mathrm{V}})}$$

**Posterior definition:** Define the posterior probability

$$\gamma(t,l) = \frac{\alpha(t,l)\beta(t,l)}{p(\mathbf{Z}^{\mathrm{V}}|\mathbf{Y}^{\mathrm{V}})}$$

Therefore, the final result is

$$\frac{\partial \mathcal{L}_{\mathrm{VCM}}}{\partial \mathbf{y}_t^{\mathrm{V}}(\mathbf{z}_l^{\mathrm{V}})} = p(\mathbf{z}_l^{\mathrm{V}}|\mathbf{y}_t^{\mathrm{V}}) - \gamma(t,l)$$

This concludes the proof. $\qquad\qquad\qquad\qquad\qquad\qquad\qquad\qquad\qquad\qquad\qquad$ $\square$

## D VCM Efficiency Analysis Based on Floating-Point Operations (FLOPs)

Evaluating the computational complexity of LVLMs requires examining key components such as the self-attention mechanism and the feed-forward network (FFN). Specifically, FLOPs can be expressed as

$$\mathrm{FLOPs} = T \times (4nd^2 + 2n^2d + 2ndm), \tag{8}$$

where $T$ is the number of transformer layers, $n$ is the sequence length, $d$ is the hidden dimension size, and $m$ represents the intermediate size of the FFN. During training, we can estimate $\mathbb{E}[n] \approx d/4$ and $m \approx 4d$. Therefore, we can approximate that $\mathrm{FLOPs} \approx T \times (12nd^2 + 2n^2d)$. We can see that the computational complexity is strongly influenced by the sequence length $n$. In typical vision-language tasks, the sequence length is defined as

$$n = n_{\mathrm{sys}} + n_{\mathrm{img}} + n_{\mathrm{ins}} + n_{\mathrm{res}}, \tag{9}$$

where $n_{\mathrm{sys}}$, $n_{\mathrm{img}}$, $n_{\mathrm{ins}}$, and $n_{\mathrm{res}}$ denote the lengths of the system prompt, vision tokens, instruction, and response, respectively. Note that $n_{\mathrm{img}}$ is often much larger than the other three, sometimes by a factor of 20 [16, 19] as $n_{\mathrm{img}} \approx 20 \times (n_{\mathrm{sys}} + n_{\mathrm{ins}} + n_{\mathrm{res}})$. Therefore, reducing $n_{\mathrm{img}}$ is essential for improving the efficiency of LVLMs. Here, we analyze the expectations of FLOPs under two scenarios. (1) The original case where $n$ is unchanged. (2) The case where $n$ is scaled by $1/8$ by VCM with information domain $S = 1/4$. Let the variance of $n$ be $\sigma_n^2$.

**Case (1).** Based on the above analysis, we first have $\mathbb{E}[n] = d/4$ and $\mathbb{E}[n^2] = \sigma_n^2 + d^2/16$. Afterward, we can derive

$$\mathbb{E}_{(1)} = T \times \left(12d^2 \cdot \mathbb{E}[n] + 2d \cdot \mathbb{E}[n^2]\right) = T \times \left(25d^3/8 + 2d\sigma_n^2\right). \tag{10}$$

**Case (2).** When $n$ is scaled by $1/8$, the new expectation is

$$\mathbb{E}_{(2)} = T \times \left(12d^2 \cdot \mathbb{E}[n'] + 2d \cdot \mathbb{E}[n'^2]\right), \tag{11}$$

where $\mathbb{E}[n'] = d/32$ and $\mathbb{E}[n'^2] = \sigma_n^2/64 + d^2/1024$. Substituting these values gives

$$\mathbb{E}_{(2)} = T \times \left(3d^3/8 + d\sigma_n^2/32 + d^3/512\right). \tag{12}$$

**Ratio about expectations of FLOPs.** The ratio of the scaled expectation (case (2)) to the original expectation (case (1)) can be approximated by discarding negligible terms as follows:

$$R = \frac{\mathbb{E}_{(2)}}{\mathbb{E}_{(1)}} = \frac{T \times \left(3d^3/8 + d\sigma_n^2/32 + d^3/512\right)}{T \times (25d^3/8 + 2d\sigma_n^2)} \approx \frac{3d^3/8}{25d^3/8} = \frac{3}{25}. \tag{13}$$

This shows that VCM can significantly reduce computational costs, *e.g.*, achieving an 85% reduction with respect to FLOPs for LLaVA-1.5-7B.

Table 5: Overall descriptions of the evaluation benchmarks for visual question answering (VQA), zero-shot image classification (ZERO.), open-vocabulary object detection (OV-OD), open-vocabulary semantic segmentation (OV-SS), and video understanding (VIDEO.).

| Tasks | Datasets | Descriptions | Eval Splits | Metrics |
|---|---|---|---|---|
| VQA | VQA$^{v2}$ [21] | Scene understanding QA | test-dev | VQA Acc (↑) [79] |
| | GQA [22] | Scene understanding QA | test-dev | VQA Acc (↑) [79] |
| | VizWiz [80] | Scene understanding QA | test-dev | VQA Acc (↑) [79] |
| | SciQA [24] | External knowledge QA | val | VQA Acc (↑) [79] |
| | POPE [25] | Visual hallucination | val | Acc (↑) |
| | MME [26] | External knowledge QA | val | VQA Acc (↑) [79] |
| | MMB [27] | Visual comprehension | dev-en | Acc (↑) |
| | SEED [28] | Visual comprehension | val | Acc (↑) |
| | MMVet [29] | Visual comprehension | val | Acc (↑) |
| | VQA$^T$ [30] | Text understanding QA | val | VQA Acc (↑) [79] |
| | MMstar [31] | Knowledge leakage QA | val | VQA Acc (↑) [79] |
| | RefCOCO [32] | Region-level VQA | val, testA, testB | CIDEr (↑) [81] |
| Zero-Shot. | COCO Panoptic [49] | Zero-shot image classification | val | Acc (↑) |
| OV-OD | OV-COCO [34] | Open-vocabulary object detection | val | AP$_{50}^{novel}$,AP$_{50}^{base}$,AP$_{50}$,(↑) [34] |
| OV-SS | ADE20K [35] | Open-vocabulary semantic segmentation | val | mIoU, mAcc (↑) [82] |
| VIDEO. | TGIF-QA [36] | Video understanding | test | Acc (↑) |
| | MSVD-QA [37] | Video understanding | test | Acc (↑) |
| | MSRVTT-QA [37] | Video understanding | test | Acc (↑) |
| | ActivityNet-QA [38] | Video understanding | test | Acc (↑) |

# E  Additional Implementation Details

**More details of evaluation.** In this paper, we demonstrate the superiority of VCM on several tasks, including visual question answering, zero-shot image classification, open-vocabulary object detection, open-vocabulary semantic segmentation, and video understanding. All these evaluation tasks and metrics are listed in Table 5.

Table 6: Side-by-side display of the original probabilities, $\alpha$ table, and $\gamma$ table (rounded to three decimal places) in dynamic programming.

| $t$ | **P** | | $\alpha$ | | | | | $\gamma$ | | | | | |
|---|---|---|---|---|---|---|---|---|---|---|---|---|---|
| | $P(\emptyset)$ | $P(\star)$ | $l=1$ | $l=2$ | $l=3$ | $l=4$ | $l=5$ | $l=1$ | $l=2$ | $l=3$ | $l=4$ | $l=5$ | total |
| 1 | 0.800 | 0.200 | 0.800 | 0.200 | 0 | 0 | 0 | 1.818 | 0.196 | 0 | 0 | 0 | 2.014 |
| 2 | 0.600 | 0.400 | 0.480 | 0.400 | 0.120 | 0 | 0 | 1.163 | 0.819 | 0.032 | 0 | 0 | 2.014 |
| 3 | 0.200 | 0.800 | 0.096 | 0.704 | 0.104 | 0.096 | 0 | 0.248 | 1.677 | 0.074 | 0.015 | 0 | 2.014 |
| 4 | 0.300 | 0.700 | 0.029 | 0.560 | 0.242 | 0.140 | 0.029 | 0.075 | 1.447 | 0.464 | 0.027 | 0.002 | 2.014 |
| 5 | 0.700 | 0.300 | 0.020 | 0.177 | 0.562 | 0.115 | 0.118 | 0.052 | 0.458 | 1.449 | 0.042 | 0.014 | 2.014 |
| 6 | 0.900 | 0.100 | 0.018 | 0.020 | 0.664 | 0.068 | 0.210 | 0.047 | 0.051 | 1.723 | 0.167 | 0.027 | 2.014 |
| 7 | 0.100 | 0.900 | 0.002 | 0.034 | 0.068 | 0.659 | 0.028 | 0.005 | 0.088 | 0.177 | 1.708 | 0.036 | 2.014 |
| 8 | 0.300 | 0.700 | 0.001 | 0.025 | 0.031 | 0.509 | 0.206 | 0.002 | 0.046 | 0.133 | 0.943 | 0.890 | 2.014 |

**More details of VCM.** To further elaborate on the details of our method, we provide a comprehensive overview of the process described in Section 3, along with the complete pseudocode for the training pipeline in Algorithm 1. The VCM loss is derived through vision concept length estimation, objective expansion, forward and backward initialization, and state transitions. Gradients are then computed to enable optimization.

Furthermore, to further clarify the dynamic programming optimization process of VCM, we present the complete procedure of computation, optimization, and inference for VCM when the input vision sequence length is 8 and the output vision concept length is 2. As shown in Table 6, the forward probabilities $\alpha$ are strictly optimized according to the recursive formulas described in the main text. Based on these, we compute the $\gamma$ table, where the sum of each row remains constant, representing the total probability of all reasonable alignment paths discussed in the main text. Moreover, we provide a corresponding visualized case in Figure 6 to help readers better understand the training and inference processes. During training, we maximize all feasible alignment paths (blue) and minimize

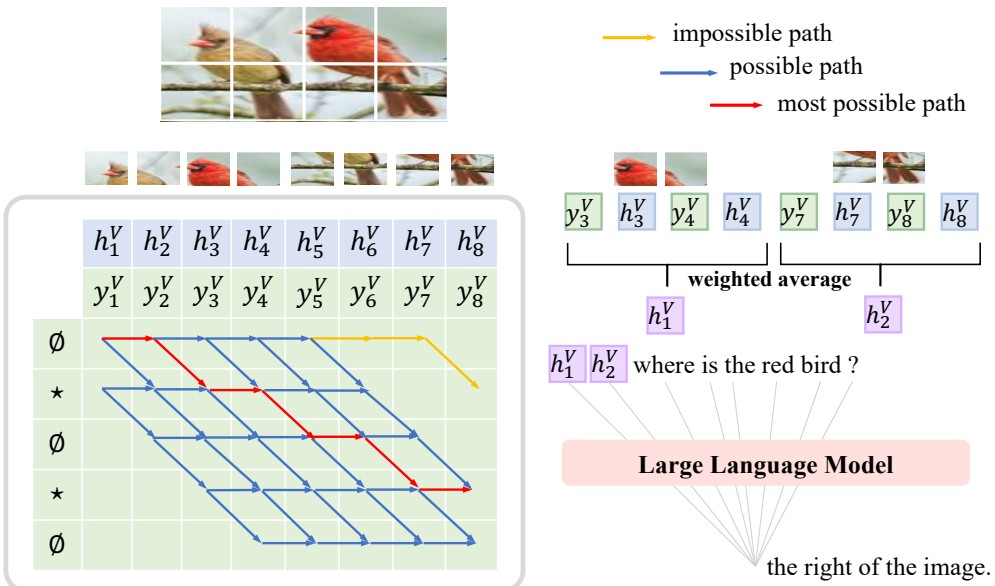

Figure 6: **An example of the workflow in VCM.** Specifically, the VCM loss computes all possible alignment paths with the extended target sequence, maximizing the probabilities of feasible paths (blue paths) while minimizing the probabilities of infeasible paths (yellow paths), which ensures that the output concept length meets expectations. During training and inference, the most probable path (red path) is selected, and a segment merging (SM) operation is performed, where adjacent retained vision features are weighted and averaged based on their probability scores. The extracted vision concept features are then combined with the instruction and fed into the LLM to generate the corresponding answer.

all infeasible alignment paths (yellow). During inference, we directly select the most probable path red among all possible alignment paths as the optimal path.

**More details of SM.** To extract vision concepts from the most probable path, we employ the segment merging (SM) operation. However, since the length and number of vision concepts vary across samples, using a double for-loop for feature extraction introduces a significant computational bottleneck, severely reducing training efficiency. This limitation is one of the primary reasons previous methods [19, 15, 16, 14] avoid using variable-length features during training.

To address this issue, we implement a parallelized SM operation capable of simultaneously handling multi-end features of varying lengths and positions. The PyTorch implementation of this operation is provided in Algorithm 2. Compared to the double for-loop approach, our method achieves a 100× speedup, making the SM operation computationally efficient and effectively imperceptible.

# F  Supplementary Experiments Results

**Results on VQA for high-resolution scenarios.** To further demonstrate the generalization capabilities of VCM, we apply it to LLaVA-NeXT [1], which supports high-resolution scenarios. Compared to LLaVA-1.5, LLaVA-NeXT divides the image into four parts, resizes the original image, and converts it into five independent images. The vision encoder processes each image to extract vision tokens, which are then merged. While this method improves model performance, it significantly increases the number of vision tokens. As shown in Table 7, our proposed VCM consistently maintains strong performance under high-resolution settings. Specifically, when reducing the number of vision tokens to only almost 5% of the original one (160 vs. 2880), our method achieves more competitive performance than SparseVLM, PDrop, and VisionZip. Furthermore, VCM demonstrates

Table 7: Performance on 6 image-based VQA benchmarks under high resolution conditions. The best result in each case is shown in bold.

| Methods | #Vision Tokens | VQA$^{v2}$ | GQA | SciQA | POPE | MME | MMB | Avg. (%) |
|---|---|---|---|---|---|---|---|---|
| LLaVA-NEXT [1] | 2880 | 80.1 | 64.2 | 70.2 | 86.5 | 1842.1 | 67.9 | 72.4 |
| SparseVLM [15] | 160 | 66.3 | 51.2 | 67.5 | 74.8 | 1542.1 | 63.1 | 63.0 |
| PDrop [48] | 160 | 69.7 | 53.9 | 67.8 | 83.1 | 1587.4 | 64.8 | 65.9 |
| VisionZip [16] | 160 | 71.4 | 55.5 | 68.3 | 83.4 | 1630.2 | 60.1 | 66.2 |
| VCM | 160 | **76.8** | **60.1** | **69.6** | **87.1** | **1723.4** | **65.5** | **70.1** |

Table 8: Performance on 4 video understanding benchmarks. The best result in each case is shown in bold.

| Methods | #Vision Tokens | TGIF-QA | MSVD-QA | MSRVTT-QA | ActivityNet-QA | Avg. (%) |
|---|---|---|---|---|---|---|
| Video-LLaVA [40] | 2048 | 47.1 | 69.8 | 56.7 | 43.1 | 54.2 |
| FastV [19] | 198 | 23.1 | 38.0 | 19.3 | 30.6 | 27.8 |
| SparseVLM [15] | 198 | 44.7 | **68.2** | 31.0 | 42.6 | 46.6 |
| VisionZip [16] | 136 | 42.4 | 63.5 | 52.1 | **43.0** | 50.3 |
| VCM | 136 | **45.4** | 66.8 | **54.9** | 42.7 | **52.5** |

outstanding performance on visual hallucination tasks (*e.g.*, POPE), validating its effectiveness in concept extraction for high-resolution images.

**Results on video understanding.** To further demonstrate the effectiveness of our method, we validate the acceleration performance of multi-frame inputs in video understanding scenarios on Video-LLaVA [40]. As shown in Table 8, compared with other token reduction methods [48, 15, 16], our method achieves better performance at the same or smaller vision token number, which verifies that our VCM exhibits stronger vision representation capabilities.

**Generality and scalability of VCM.** To explore whether VCM can achieve further performance gains through increased data and model scale, we conduct experiments using a larger dataset. We increase training steps and use a larger language model (Vicuna-13B) under the default configuration. As shown in Table 9, increasing the amount of instruction-tuning data and scaling up the language model both improve performance, which demonstrates the strong scalability of our VCM.

In addition, we conduct experiments on the QwenVL architecture to verify the architectural generalization ability of VCM. Specifically, to ensure fairness, we use fully open-source LLaVA data [20] and load the official weights of Qwen2VL [2] for pre-training and instruction fine-tuning. As shown in Table 9, under the same number of training steps, our method significantly reduces the vision token number and accelerates inference while slightly sacrificing performance. This demonstrates that our VCM has strong generalization ability across different architectures.

**FLOP comparison.** To better demonstrate the speed improvement brought by our method, we compare the FLOPs under different configurations with other methods [19, 48, 15], as shown in Table 10. It can be observed that under various configurations, the FLOPs and latency of our method have reached the state-of-the-art levels while achieving superior performance. This is attributed to the parallelized optimization of the SM operation and the powerful representation capabilities of vision concept modeling, which highlight the superiority of VCM.

Table 9: Results of VCM generality and scalability on 4 image-based VQA benchmarks. The best result in each case is shown in bold.

| Methods | LLMs | Training Steps | #Vision Tokens | SciQA | POPE | VQA$^{v2}$ | GQA | Avg. (%) |
|---|---|---|---|---|---|---|---|---|
| Qwen2-VL | Qwen2-7b | 500 | 1326 | 73.5 | **80.2** | **69.8** | 57.8 | **70.3** |
| Qwen2-VL+VCM | Qwen2-7b | 500 | 576 | **74.1** | 79.2 | 68.3 | 56.6 | 69.6 |
| LLaVA-v1.5+VCM | Vicuna-7b | 500 | 144 | 64.1 | 76.6 | 56.8 | 43.6 | 60.3 |
| LLaVA-v1.5+VCM | Vicuna-7b | 1000 | 144 | 65.4 | 78.7 | 59.3 | 46.1 | 62.4 |
| LLaVA-v1.5+VCM | Vicuna-13b | 1000 | 144 | **67.8** | **80.3** | **62.3** | **49.5** | **64.9** |

Table 10: Performance of VCM under different vision token configurations. In addition to the theoretical analysis of FLOPs, we also provide a comparison of performance, FLOPs, and latency under different vision token lengths in practical scenarios.

| Methods | #Vision Tokens | FLOPs (T) | Latency (ms) | GQA | MMB | MME | POPE | SciQA | SEED | VQA$^{T}$ | MMVet | Avg. (%) |
|---|---|---|---|---|---|---|---|---|---|---|---|---|
| LLaVA-v1.5 [20] | 576 | 4.62 | 57.82 | 62.0 | 64.3 | 1862.1 | 85.9 | 66.8 | 58.6 | 58.2 | 30.5 | 61.6 |
| FastV [19] | 128 | 1.70 | 30.70 | 49.6 | 56.1 | 1490.0 | 53.4 | 68.6 | 48.1 | 50.5 | 26.3 | 50.7 |
| PDrop [48] | 128 | 1.62 | 37.77 | 56.0 | 61.1 | 1664.2 | 82.3 | 69.9 | 53.3 | 55.1 | 30.8 | 58.5 |
| SparseVLM [15] | 128 | 1.72 | 33.28 | 58.4 | 64.5 | 1746.1 | 85.0 | 68.6 | 58.2 | 56.7 | 29.0 | 60.3 |
| VCM | 128 | 1.71 | 31.42 | 61.2 | 63.7 | 1789.2 | 86.1 | 68.5 | 62.7 | 56.2 | 33.4 | 62.0 |
| FastV [19] | 64 | 1.29 | 27.30 | 46.1 | 47.2 | 1255.4 | 38.2 | 68.7 | 43.7 | 47.8 | 19.6 | 44.5 |
| PDrop [48] | 64 | 1.18 | 43.41 | 41.9 | 33.3 | 1092.3 | 55.9 | 69.2 | 40.0 | 45.9 | 30.7 | 44.5 |
| SparseVLM [15] | 64 | 1.30 | 29.89 | 53.8 | 60.1 | 1589.2 | 77.5 | 69.8 | 52.2 | 53.4 | 24.9 | 56.1 |
| VCM | 64 | 1.24 | 28.46 | 60.8 | 63.4 | 1719.6 | 85.2 | 69.1 | 61.4 | 54.8 | 31.2 | 60.9 |

## G More Visualization Results

To further demonstrate the effectiveness of our method, we provide additional visualization results. Specifically, they include vision concept extraction in Figure 7, vision concept enhancement of the vision encoder in Figure 8, open-vocabulary object detection in Figure 9, and open-vocabulary semantic segmentation in Figure 10. Overall, these visualization results provide strong evidence for the superiority of our VCM.

## H Broader Impacts

The proposed VCM framework advances the efficiency and applicability of LVLMs. It obviously reduces computational costs while enhancing semantic alignment between vision and text, enabling scalable deployment in resource-constrained environments such as edge devices and mobile platforms. Furthermore, VCM's fine-grained concept extraction has the potential to improve performance in critical applications like autonomous driving and medical imaging, where efficient and precise visual understanding is vital. Nevertheless, its deployment must account for possible biases inherited from pre-trained models. Ensuring responsible AI usage remains a key consideration for broader societal impact. Overall, VCM represents a step forward in sustainable and accessible vision-language intelligence, paving the way for efficient multimodal learning across diverse real-world scenarios.

## I Limitations

In our experiments, we primarily rely on an adaptive keyword selection strategy. However, this way may introduce certain biases, as the selected keywords may not always represent true keywords. Additionally, to simplify modeling, we use min-max normalization for estimated vision concept length. This coarse-grained design may negatively impact the model's performance. In the future, we aim to employ open-source language models to perform more refined keyword selection for instruction data. By providing finer-grained additional information, we hope to further improve the performance of the proposed method.

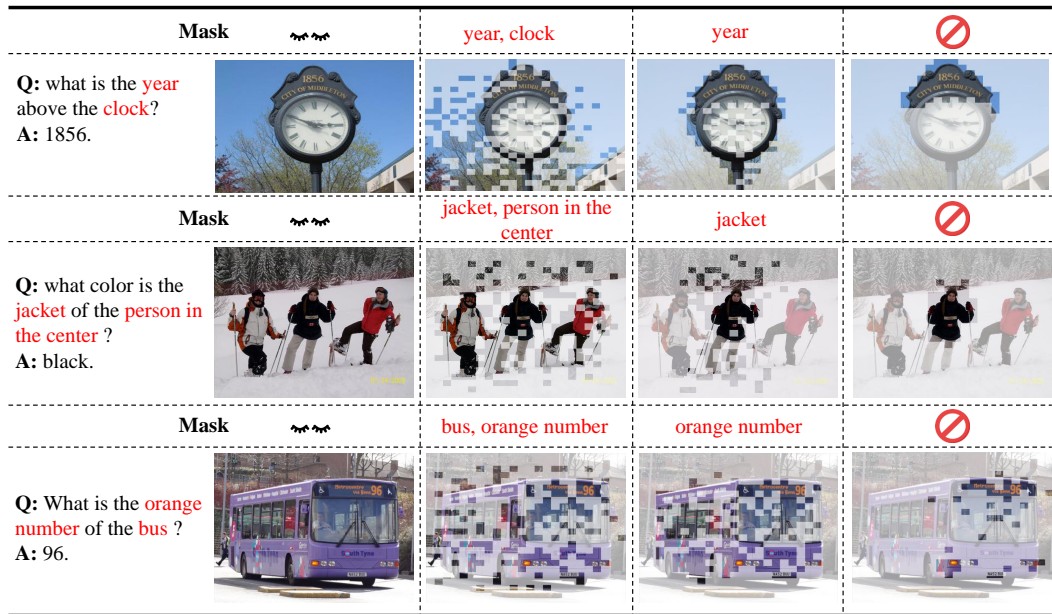

Figure 7: **More visualization results of VCM on different instructions.** From left to right, the visual representation becomes increasingly sparse, leaving corresponding vision tokens to unmasked keywords (highlighted in red). VCM can capture the corresponding vision concepts relatively accurately by masking different numbers of keywords across different instructions, which demonstrates its robustness and effectiveness.

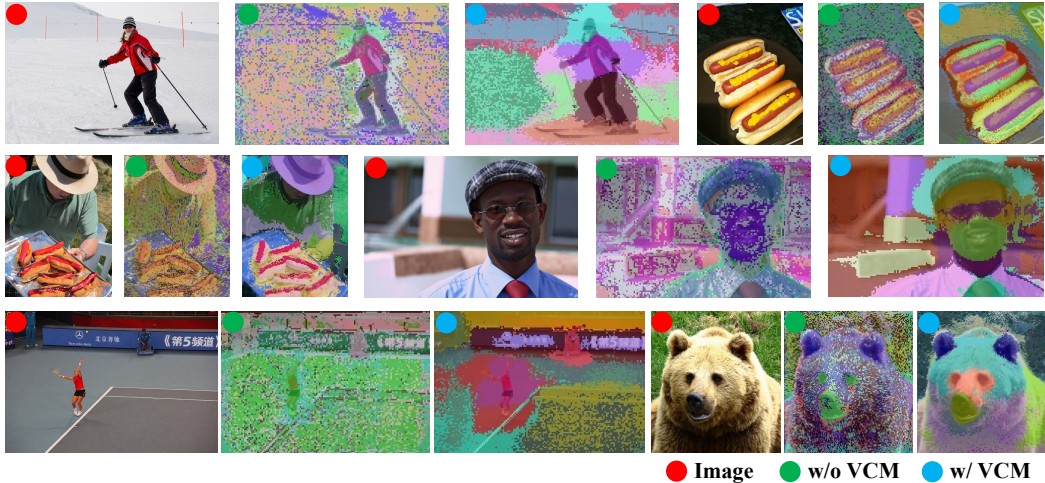

Figure 8: **More K-Means visualization of the dense feature maps of CLIP ViT trained with VCM.** Specifically, we show the raw images, the K-Means results without VCM, and those of our fine-tuned model by VCM. As we can see, VCM has significant conceptual enhancements in multiple scenarios, such as animals, food, humans, and near/far perspectives, proving the generalizability and potential of our method.

**Algorithm 1** Pseudocode for VCM Loss and Gradient Computation

---

**Input:**

Logit sequence $\mathbf{Y}^{\mathrm{V}} = [\mathbf{y}_1^{\mathrm{V}}, \mathbf{y}_2^{\mathrm{V}}, \ldots, \mathbf{y}_M^{\mathrm{V}}]$, where $M$ is the input sequence length.

Information domain scalar $S \in (0, 1/2]$ for length adjustment.

Mask ratio $r \sim \mathrm{Uniform}(0, 1)$.

Keyword count prior $N_{\text{text}}$.

**Output:**

VCM loss $\mathcal{L}_{\mathrm{VCM}}$ and gradients $\frac{\partial \mathcal{L}_{\mathrm{VCM}}}{\partial \mathbf{y}^{\mathrm{V}}}$.

**Initialization:**

Set $L = \lfloor M \cdot S \cdot (1 - \mathrm{Norm}(N_{\text{text}})) \rfloor$ as the length of the target sequence.

Sample target sequence $\mathbf{x}$ with $L$ symbols:

$$\mathbf{Z}^{\mathrm{V}} = [\mathbf{z}_1^{\mathrm{V}}, \mathbf{z}_2^{\mathrm{V}}, \ldots, \mathbf{z}_L^{\mathrm{V}}], \quad \mathbf{z}_l^{\mathrm{V}} \in \{\star\}$$

Here, $\star$ denotes retaining the token, and $\emptyset$ denotes a blank.

Interleave $\emptyset$ between and around the symbols:

$$\mathbf{Z}^{\mathrm{V}} = [\emptyset, \mathbf{z}_1^{\mathrm{V}}, \emptyset, \mathbf{z}_2^{\mathrm{V}}, \emptyset, \ldots, \mathbf{z}_L^{\mathrm{V}}, \emptyset]$$

The resulting length of $\mathbf{Z}^{\mathrm{V}}$ is $2L + 1$.

Initialize all forward variables $\alpha(t, l)$ and backward variables $\beta(t, l)$ to 0.

Set $\alpha(1, 1) = p(\emptyset | \mathbf{y}_1^{\mathrm{V}})$, $\alpha(1, 2) = p(\star | \mathbf{y}_1^{\mathrm{V}})$

$\beta(M, 2L + 1) = p(\emptyset | \mathbf{y}_M^{\mathrm{V}})$ and $\beta(M, 2L) = p(\star | \mathbf{y}_M^{\mathrm{V}})$.

**Forward Pass:**

**for** $t = 2$ **to** $M$ **do**

  **for** $l = 1$ **to** $2L + 1$ **do**

    $\alpha(t, l) = p(\mathbf{z}_l^{\mathrm{V}} | \mathbf{y}_t^{\mathrm{V}}) \cdot (\alpha(t - 1, l) + \alpha(t - 1, l - 1))$

  **end for**

**end for**

**Backward Pass:**

**for** $t = M - 1$ **to** $1$ **do**

  **for** $l = 1$ **to** $2L + 1$ **do**

    $\beta(t, l) = (\beta(t + 1, l) + \beta(t + 1, l + 1)) \cdot p(\mathbf{z}_l^{\mathrm{V}} | \mathbf{y}_t^{\mathrm{V}})$

  **end for**

**end for**

**Compute Loss:**

  $p(\mathbf{Z}^{\mathrm{V}} | \mathbf{Y}^{\mathrm{V}}) = \sum_{l=1}^{2L+1} \alpha(M, l) \cdot \beta(M, l)$

  $\mathcal{L}_{\mathrm{VCM}} = -\log p(\mathbf{Z}^{\mathrm{V}} | \mathbf{Y}^{\mathrm{V}})$

**Compute Gradients:**

**for** $t = 1$ **to** $M$ **do**

  **for** $l = 1$ **to** $2L + 1$ **do**

    Compute posterior probability:

$$\gamma(t, l) = \frac{\alpha(t, l) \cdot \beta(t, l)}{p(\mathbf{Z}^{\mathrm{V}} | \mathbf{Y}^{\mathrm{V}})}$$

    Compute the gradient for $\mathbf{y}_t^{\mathrm{V}}(\mathbf{z}_l^{\mathrm{V}})$:

$$\frac{\partial \mathcal{L}_{\mathrm{VCM}}}{\partial \mathbf{y}_t^{\mathrm{V}}(\mathbf{z}_l^{\mathrm{V}})} = p(\mathbf{z}_l^{\mathrm{V}} | \mathbf{y}_t^{\mathrm{V}}) - \gamma(t, l)$$

  **end for**

**end for**

**Optimization:**

Update $\mathbf{y}_t^{\mathrm{V}}$ using gradient descent:

$$\mathbf{y}_t^{\mathrm{V}}(\mathbf{z}_l^{\mathrm{V}}) \leftarrow \mathbf{y}_t^{\mathrm{V}}(\mathbf{z}_l^{\mathrm{V}}) - \eta \cdot \frac{\partial \mathcal{L}_{\mathrm{VCM}}}{\partial \mathbf{y}_t^{\mathrm{V}}(\mathbf{z}_l^{\mathrm{V}})},$$

where $\eta$ is the learning rate.

---

**Algorithm 2** Pseudocode of Merging Segments with Scores (**100× faster than a double loop**)

```python
def merge_segments_with_scores(x, x_mask, scores):
    """
    Merge adjacent features with mask=1 using score-weighted averaging,
    discarding features with mask=0.
    Args:
        x (torch.Tensor): Input features, shape [B, S, D].
        x_mask (torch.Tensor): Binary mask, shape [B, S].
        scores (torch.Tensor): Scores for each feature, shape [B, S].
    Returns:
        new_x (list[torch.Tensor]): Merged features, each element shape [N,
     D].
    """
    B, S, D = x.size()
    # Mask features and scores
    masked_x = x * x_mask.unsqueeze(-1)
    masked_scores = scores * x_mask
    # Extend mask and scores for cumulative sums
    expanded_x_mask = torch.cat([x_mask, torch.zeros((B, 1), dtype=x_mask.
    dtype, device=x_mask.device)], dim=1)
    expanded_scores = torch.cat([scores, torch.zeros((B, 1), dtype=scores.
    dtype, device=scores.device)], dim=1)
    # Compute cumulative sums
    cumsum_mask = torch.cumsum(expanded_x_mask, dim=1)
    cumsum_x = torch.cumsum(masked_x * scores.unsqueeze(-1), dim=1)  #
    Weighted sum of features
    cumsum_scores = torch.cumsum(masked_scores, dim=1)  # Cumulative sum of
     scores
    # Identify segment boundaries
    segment_mask = x_mask * ((cumsum_mask[:, 1:] - cumsum_mask[:, :-1]) ==
    0)
    new_x = []
    for i in range(B):
        # Extract cumulative sums and segment masks for the current sample
        cumsum_x_current = cumsum_x[i]
        cumsum_scores_current = cumsum_scores[i]
        cumsum_mask_current = cumsum_mask[i][:-1]
        segment_mask_current = segment_mask[i].bool()
        # Get segment-wise cumulative sums
        segment_sums = cumsum_x_current[segment_mask_current, :]  # [N, D]
        segment_score_sums = cumsum_scores_current[segment_mask_current]  #
     [N]
        # Extend cumulative sums for segment-wise computation
        expand_segment_sums = torch.cat([torch.zeros((1, D), dtype=
    segment_sums.dtype, device=segment_sums.device), segment_sums])
        expand_segment_score_sums = torch.cat([torch.zeros((1,), dtype=
    segment_score_sums.dtype, device=segment_score_sums.device),
    segment_score_sums])
        diff_sums = expand_segment_sums[1:, :] - expand_segment_sums[-1,
    :]  # [N, D]
        diff_score_sums = expand_segment_score_sums[1:] -
    expand_segment_score_sums[:-1]  # [N]
        # Prevent division by zero
        diff_score_sums = torch.where(diff_score_sums == 0, torch.ones_like
    (diff_score_sums), diff_score_sums)
        # Compute score-weighted mean
        weighted_mean_segments = diff_sums / diff_score_sums.unsqueeze(-1)
     # [N, D]
        new_x.append(weighted_mean_segments)
    return new_x
```

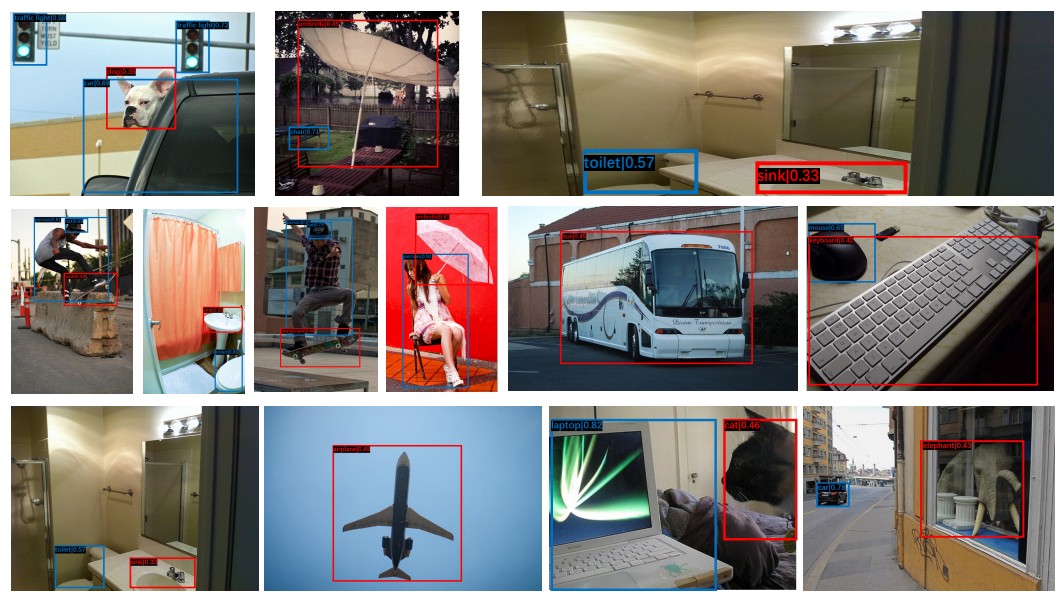

Figure 9: **Visualization of open-vocabulary object detection results.** The red boxes are for novel categories. The blue boxes are for base categories.

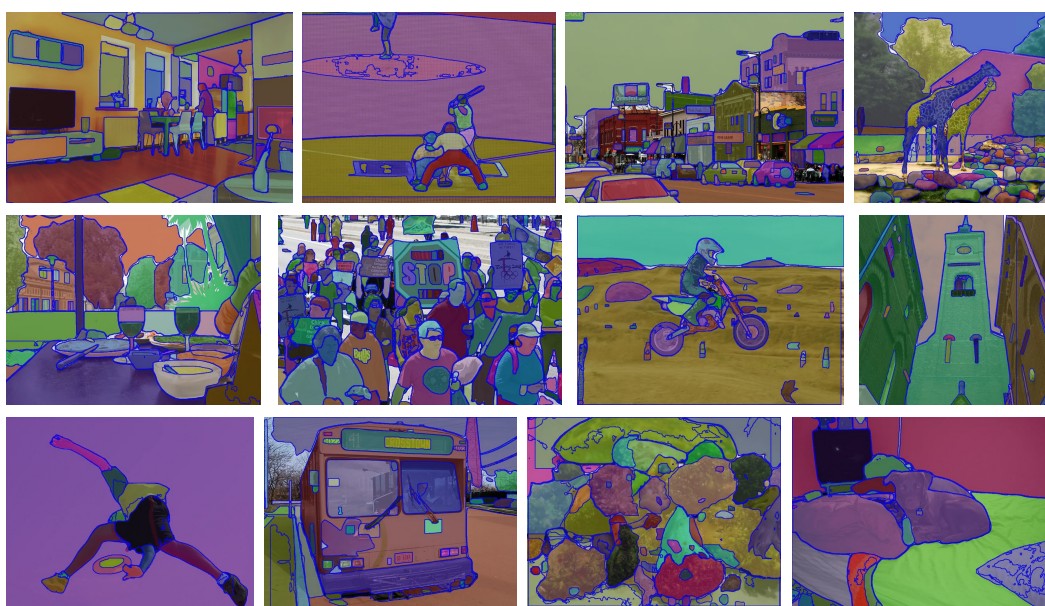

Figure 10: **Visualization of open-vocabulary semantic segmentation.** The images are sampled from the COCO val set [33].

