# OpenReview forum: "VCM: Vision Concept Modeling with Adaptive Vision Token Compression via Instruction Fine-Tuning"
_NeurIPS.cc/2025/Conference — NeurIPS 2025 poster_

### Official Review · Reviewer_RMxa · 2025-07-01

**Clarity:** 3
**Significance:** 3
**Originality:** 3
**Rating:** 4
**Confidence:** 3

**Summary:**

This paper addresses the computational inefficiency of Large Vision-Language Models (LVLMs) which typically process all visual tokens from an image regardless of the specific query. The authors introduce Vision Concept Modeling (VCM), a novel self-supervised framework that allows a model to dynamically select a sparse, relevant set of "vision concepts" based on the user's textual instructions. At its core, VCM uses a forward-backward optimization algorithm to learn this adaptive token selection without requiring expensive, fine-grained annotations. This approach is shown to drastically reduce computational costs, achieving up to an 85% reduction in FLOPs for LLaVA-1.5-7B, while maintaining strong performance across a wide range of vision-language tasks.

**Questions:**

My questions aim to clarify two counter-intuitive experimental results:

1.  **Ablation Anomaly (Table 4):** The ablation study shows that using 36 vision tokens surprisingly outperforms using 72 tokens. Could the authors provide a more detailed explanation for this counter-intuitive result? A convincing analysis would strengthen my confidence in the method's robustness.

2.  **Fine-tuning Impact (Table 1):** Fine-tuning the vision encoder (`VCM*`) led to slightly lower average performance than keeping it frozen (`VCM`), which is unusual. Could the authors clarify the potential reasons for this, such as optimization instability? Addressing this surprising result would demonstrate a more thorough experimental analysis.

**Ethical Concerns:**

["NO or VERY MINOR ethics concerns only"]

**Final Justification:**

Thank you for the thorough and well-prepared rebuttal. Your responses, supported by new experimental results, have successfully addressed all of my initial concerns.

Specifically:

* The new ablations on heuristic length (W1) and keyword selection robustness (W2) are very convincing.

* The latency breakdown (W3) provides the missing analysis on VCM's overhead.

* Your explanations for the counter-intuitive results in the ablation (Q1) and vision encoder fine-tuning (Q2) were particularly insightful, clarifying that these are known phenomena consistent with prior work and community best practices.

Given the strength of this rebuttal, my concerns have been resolved. I am now happy to support the paper and will raise my rating accordingly.

**Quality:**

3

**Strengths And Weaknesses:**

### **Strengths**

* **Broad Applicability and Enhanced Capabilities:** The method's effectiveness extends beyond standard VQA, significantly enhancing the performance of dense perception models. For instance, it improves the $AP_{50}$ score in open-vocabulary object detection and boosts performance in open-vocabulary semantic segmentation across multiple datasets.

* **Novel and Principled Methodology:** The use of a forward-backward algorithm based on dynamic programming is a technically sound and original approach to selecting a variable number of vision tokens. This provides an efficient solution ($\mathcal{O}(M^2)$ complexity) to an otherwise intractable search problem, all without requiring expensive, fine-grained annotations.

### **Weaknesses**

* **Heuristic-Based Concept Length Estimation:** The model relies on a fixed, heuristic formula to estimate the required number of vision tokens (`L`). This length is calculated using min-max normalization of keyword counts, not learned adaptively, which the authors acknowledge may negatively impact performance.

* **High Dependency on Keyword Selection:** The framework's performance is critically dependent on the accuracy of the keyword selection module. The paper concedes that this process can introduce biases, meaning an inaccurate keyword selection could compromise the entire vision concept extraction process.

* **Lack of Analysis on VCM's Own Overhead:** The efficiency analysis primarily highlights the FLOPs reduction within the LLM itself. The paper does not provide a detailed breakdown of the computational overhead introduced by the VCM components (e.g., keyword selection and dynamic programming), which must run before the LLM processes the tokens.

* **High Architectural Complexity:** The VCM framework is architecturally complex, integrating multiple distinct components such as a keyword selector, a cross-attention projector, and the dynamic programming module. This complexity could pose a barrier to easy implementation and adoption when compared to simpler token reduction methods.

* **Computational Analysis Deferred to Appendix:** While the paper claims significant computational savings, the detailed theoretical analysis of FLOPs is located in the appendix rather than the main paper. Including a more direct summary of this analysis in the main body would better substantiate the efficiency claims upfront.

I am willing to raise my rating if the authors can satisfactorily address these questions in their rebuttal.

---

> ### Author Rebuttal · Authors · 2025-07-31
>
> Thanks for your professional and careful review. We respond to your concerns or questions as follows.
>
> > **W1**: Heuristic-Based Concept Length Estimation.
>
> **Response:**
>
>
> To investigate the impact of estimated length versus precise length on performance, we conducted the following comparative ablation experiments. Specifically, we trained the model for 500 steps using 5K samples (as mentioned in line 85 of the paper), which were labeled by GPT-4o.
>
> | length      | Token | SciQA    | VizWiz   | POPE     | MME        |
> | :---------- | :---- | :------- | :------- | -------- | ---------- |
> | estimated   | 144   | **63.5** | **47.2** | 76.2     | 1334.4     |
> | api-labeled | 144   | 62.9     | 47.1     | **76.4** | **1337.2** |
>
> ------
>
> As shown in the table, the experimental results under precise length and estimated length settings are quite similar. This is attributed to VCM's inherent error-tolerance mechanism:
>
> 1) **Handling underestimated lengths**: When the estimated length is shorter than ideal, VCM compensates by merging more vision tokens into a single vision concept token, thereby increasing the average information capacity of each vision concept token.
>
> 2) **Handling overestimated lengths**: When the estimated length is longer than necessary, VCM generates longer vision concept tokens, ensuring that the subsequent LLM can still extract sufficient information for downstream tasks.
>
> > **W2**: High Dependency on Keyword Selection.
>
> **Response:**
> The keyword selection module in VCM demonstrates reliable performance in selecting relevant keywords. Additionally, VCM's random masking strategy introduces error tolerance and robustness against incorrect keyword predictions. Together, these two components ensure efficient and accurate learning, thereby reducing the dependency on keyword selection. To separately validate these two aspects, we conducted the following experiments:
>
> To better illustrate the reliability of the keyword selection module, we randomly selected 1K samples from the 5K samples (as mentioned in line 85 of the paper) labeled by GPT-4o and computed the success rate of keyword selection by the module during training. The experimental results are reported in the table below.
>
> | Training Steps | 0.5k | 1k   | 1.5k | 2k   | 2.5k | 3k   | 3.5k | 4k   | 4.5k | 5k   |
> | :------------- | ---- | :--- | :--- | :--- | ---- | ---- | ---- | ---- | ---- | ---- |
> | Accuracy       | 19.4 | 48.9 | 67.4 | 76.5 | 83.6 | 88.9 | 91.3 | 92.6 | 94.1 | 94.3 |
>
> As shown in the table, during pretraining, the keyword selection module progressively learns the relationships between images and text, ultimately achieving a 94.3% success rate in selecting the correct keywords.
>
> To further validate the robustness of VCM, we simulated unreliable predictions by the keyword selection module during inference. Specifically, we randomly replaced selected keywords with non-keywords at different replacement probabilities $r$, and then tested the model while recording the average visual concept token length.
>
> | r    | Mean Token | SciQA | POPE | MME    |
> | :--- | :--------- | :---- | ---- | ------ |
> | 0%   | 64         | 69.1  | 85.2 | 1719.6 |
> | 15%  | 76         | 68.9  | 85.1 | 1720.3 |
> | 30%  | 89         | 69.0  | 84.9 | 1725.4 |
> | 45%  | 96         | 68.7  | 85.0 | 1726.8 |
> | 60%  | 103        | 68.9  | 85.1 | 1728.7 |
>
> As shown in the table, VCM exhibits strong error tolerance. When the keyword selection module makes incorrect predictions, VCM tends to generate longer visual concepts to ensure that the subsequent LLM can access sufficient prior information for downstream tasks.
>
> > **W3**: Lack of Analysis on VCM's Own Overhead.
>
> **Response:**
>
> We sincerely thank the reviewers for their valuable suggestions, which have significantly improved the quality of our work. In response, we have added the following experiments to provide more detailed insights into the latency overhead of VCM components and compared them with other methods. Specifically, we separated the latency contributions of the original LLM and the additional modules introduced by each method for more precise comparisons.
>
> | Method    | Token | Not LLM(ms) | LLM(ms) | Total |
> | :-------- | :---- | :---------- | :------ | ----- |
> | LLaVA     | 576   | 0           | 57.82   | 57.82 |
> | FastV     | 128   | 2.39        | 28.31   | 30.70 |
> | PDrop     | 128   | 10.01       | 27.76   | 37.77 |
> | SparseVLM | 128   | 6.93        | 26.45   | 33.28 |
> | VCM       | 128   | 2.88        | 28.36   | 31.24 |
>
> As shown in the table, two key observations can be made. First, the non-LLM overhead introduced by VCM is only slightly higher than FastV. This is because our method utilizes a lightweight design consisting of only a two-layer attention module, resulting in no significant increase in parameter count compared to previous methods. Second, our approach incorporates weighted averaging and parallel optimization techniques, as detailed in Appendix Algorithm 2, which further reduces the latency caused by the additional components in our method.
>
> > **W4**: High Architectural Complexity.
>
> **Response:**
>
> We sincerely thank the reviewer for their careful review. In fact, while VCM may appear theoretically complex, in practice it consists of only two attention layers, resulting in no significant increase in parameter count compared to previous methods. Additionally, its computational complexity remains the same as earlier approaches, at $O(M^2)$. Moreover, we have implemented parallel optimization techniques, enabling VCM to operate efficiently.
>
> As a plug-and-play method, VCM has consistently demonstrated its simplicity, efficiency, and versatility across diverse model architectures (e.g., LLaVA [1], QwenVL [2]) and various application scenarios (e.g., standard images, high-resolution images, and video). This further highlights its generalizable nature and adaptability for different use cases.
>
> > **W5**: Computational Analysis Deferred to Appendix.
>
> **Response:**
>
> We sincerely thank the reviewer for their thoughtful suggestions, which have helped improve the readability of our work. Due to page limitations, we chose to place the performance comparisons in the appendix. In the revised version, we will include a dedicated discussion and highlight the location of the relevant tables to ensure readers can easily find and focus on them.
>
> > **Q1**: Could the authors provide a more detailed explanation for this counter-intuitive result?
>
> **Response:**
>
> Using fewer tokens yielding better performance on certain benchmarks is a normal phenomenon and has been observed multiple times in previous works, such as MQT-LLaVA [1] (Table 1, MMB), FastV [2] (Table 1, MMMU), PDrop [3] (Table 1, POPE), and VisionZip [4] (Table 11, VizWiz). This arises because different tasks demand varying levels of perceptual granularity from the model.
>
> For instance, tasks like VizWiz emphasize local perceptual abilities, where excessive tokens may introduce noise and redundant information, leading to reduced performance compared to fewer tokens. Conversely, tasks like MME rely heavily on global details, where more tokens provide greater benefits, making such phenomena less likely to occur. This difference in task requirements explains the seemingly counterintuitive results observed for certain benchmarks.
>
> > **Q2**: Fine-tuning the vision encoder led to slightly lower average performance than keeping it frozen, which is unusual.
>
> **Response:**
>
> For LLaVA and subsequent works [3,4,5,6] based on LLaVA, freezing the vision encoder has become the default configuration. This is not due to increased computational overhead—since the vision encoder's computation cost is negligible compared to that of large language models—but rather because experiments have shown that jointly training the vision encoder with low-resolution images and limited data often yields suboptimal results.
>
> A commonly accepted explanation within the community is that freezing the vision encoder helps preserve the pre-trained knowledge in the encoder, preventing its disruption when trained on small datasets. To ensure fair comparisons, we also report results for models with the vision encoder unfrozen in Table 1. As can be observed, unfreezing the vision encoder parameters indeed leads to suboptimal performance, particularly for the baseline LLaVA-v1.5 model.
>
> [1] LLaVA: Large Language and Vision Assistant
>
> [2] Qwen-VL: A Versatile Vision-Language Model for Understanding, Localization, Text Reading, and Beyond
>
> [3] MQT-LLaVA: Matryoshka Query Transformer for Large Vision-Language Models
>
> [4] An Image is Worth 1/2 Tokens After Layer 2: Plug-and-Play Inference Acceleration for Large Vision-Language Models
>
> [5] PyramidDrop: Accelerating Your Large Vision-Language Models via Pyramid Visual Redundancy Reduction
>
> [6] VisionZip: Longer is Better but Not Necessary in Vision Language Models
>
>
>
> Thank you again for your insightful comments. If you have other comments, we are happy to address them to polish this work. We look forward to contributing to the development of both the multi-modal research and the open-source community.

---

> > ### Author Response · Authors · 2025-08-04
> >
> > Dear Reviewer  RMxa,
> >
> > We hope this message finds you well. We are writing to sincerely thank you for taking the time to review our rebuttal and for providing such insightful and constructive feedback on our submission. Your expertise and attention to detail are truly appreciated and have been incredibly beneficial in guiding our research.
> >
> > The suggestions you provided have opened up new perspectives and opportunities for enhancing the quality of our work. We are committed to implementing your recommendations to the best of our ability and are eager to explore the ideas you have mentioned further.
> >
> > Thank you once again for your invaluable contribution to our research. Your support and guidance are deeply appreciated, and we are hopeful that the revisions will meet your expectations.
> >
> > Sincerely,
> >
> > Author

---

> > ### Comment · Reviewer_RMxa · 2025-08-06
> >
> > Thank you for the thorough and well-prepared rebuttal. Your responses, supported by new experimental results, have successfully addressed all of my initial concerns.
> >
> > Specifically:
> >
> > * The new ablations on heuristic length (W1) and keyword selection robustness (W2) are very convincing.
> >
> > * The latency breakdown (W3) provides the missing analysis on VCM's overhead.
> >
> > * Your explanations for the counter-intuitive results in the ablation (Q1) and vision encoder fine-tuning (Q2) were particularly insightful, clarifying that these are known phenomena consistent with prior work and community best practices.
> >
> > Given the strength of this rebuttal, my concerns have been resolved. I am now happy to support the paper and will raise my rating accordingly.

---

> > > ### Author Response · Authors · 2025-08-06
> > >
> > > We sincerely thank you for taking the time to review our paper. Your continued engagement and the valuable concerns you've highlighted are greatly appreciated. Thank you again for your time and insights! If you have other comments, we are happy to address them to polish this work. We look forward to contributing to the development of both the multi-modal research and the open-source community.

---

### Official Review · Reviewer_u1dV · 2025-07-01

**Clarity:** 2
**Significance:** 3
**Originality:** 3
**Rating:** 5
**Confidence:** 4

**Summary:**

This paper proposes the framework of vision concept modeling through a three stage pipeline: instruction prior, key word extraction and dynamic programming selection of visual concept tokens, compresses visual tokens for MLLMs. The aim is to reduce computation while maintaining multi-vision language understanding tasks. Compared with state-of-the-art approaches, VCM cuts FLOPs by up to 85%, and with just 144 tokens it matches or surpasses a 576-token LLaVA-1.5 on eleven VQA benchmarks.

**Questions:**

Q1: If we perform self-supervised contrastive learning directly on vision tokens (without using text keywords), how large is the performance gap compared to VCM? Can this comparison demonstrate the necessity of the keyword module?

Q2: What is the average VQA score of MQT-LLaVA under the 144-token condition? Does VCM's advantage stem from differences in search space or from textual priors?

Q3: Under 1024×1024 high-resolution images and 30 fps video streaming conditions, what are the comparative metrics of VCM versus VisionZip and SparseVLM in terms of GPU latency and memory consumption?

**Ethical Concerns:**

["NO or VERY MINOR ethics concerns only"]

**Final Justification:**

The authors’ rebuttal successfully resolved all of my major concerns:

- **Sensitivity of τ (W1.1)**: They ran an ablation showing that task-specific τ values do not improve performance and explained why; they also proposed a more flexible inference-time alternative.
- **Framework clarity (W2)**: They agreed to redraw the diagram to bind symbols to tokens.
- **Effectiveness of keyword module (Q1/W1.2)**: An extra experiment confirmed that removing KM hurts performance, proving its necessity.
- **Comparison with MQT-LLaVA at 144 tokens (Q2/W1.3)**: The authors provided the missing number and articulated why textual priors give VCM an edge.
- **High-res/video metrics (Q3)**: New tables show VCM is competitive in latency and memory, with clear explanations of efficiency gains.

Because these issues were convincingly addressed with data and sound reasoning, I raised my score.

**Limitations:**

yes

**Paper Formatting Concerns:**

No formatting issues

**Quality:**

3

**Strengths And Weaknesses:**

Strenghts:
- The empirical study about relationship of keywords in instruction，keywords in response and needed visual token length provides valuable insights for future work.
- Compared to previous methods，VCM can dynamically select and merge visual tokens while maintaining the original spatial order, which is critical for dense visual prediction tasks.

Weaknesses:
- Ablation study is not sufficient
  - The length estimation employs min-max normalization with a fixed domain-specific threshold τ, lacking sensitivity analysis; whether task-specific τ values are required for different tasks remains unaddressed.
  - Lack the effectiveness of direct token selection learning without a keyword module.
  - No direct comparison was made with variable-length token encoding methods (e.g., Matryoshka Query Transformer) within the same token count range.
- The figure of the overall framework is not clear enough. I suggest you can bind G^T,G^V..., with the corresponding tokens in the figure.

---

> ### Author Rebuttal · Authors · 2025-07-31
>
> Thanks for your professional and careful review. We respond to your concerns or questions as follows.
>
> > **W1.1**: The length estimation employs min-max normalization with a fixed domain-specific threshold τ, lacking sensitivity analysis; whether task-specific τ values are required for different tasks remains unaddressed.
>
> **Response:**
>
> We hypothesize that the fixed domain-specific threshold $\tau$ refers to the information domain scalar $S$. To investigate the impact of the dynamic information domain $S$ on VCM, we categorized the training data. From the results in the ablation study shown in Table 4, we observed that for localization and fine-grained perception tasks, a smaller information domain yields better results. Conversely, for tasks requiring global information, a larger information domain is more effective.
>
> Based on this observation, we categorized the instruction-tuning data by task type. For open-ended tasks, we set the information domain to $1/2$, while for fine-grained perception tasks like the RefCOCO localization task, we set the information domain to $1/4$. We then conducted mixed training for 500 steps, and the experimental results are presented below.
>
> | Method          | SciQA | VizWiz | POPE | MME    |
> | :-------------- | :---- | :----- | :--- | ------ |
> | mix-S (1/4&1/2) | 60.5  | 44.6   | 70.7 | 1288.8 |
> | fix-S （1/4）   | 64.1  | 48.2   | 76.6 | 1350.6 |
>
> As shown in the table, using mixed training with different information domains for different tasks did not lead to further performance improvements. We believe this is because dynamic information domains introduce challenges for the forward-backward optimization algorithm, making it difficult for VCM to discriminate whether the varying learning lengths are caused by differences in information domains or differences in keyword count. This conflict ultimately led to degraded performance.
>
> Compared to dynamically altering the information domain during the training phase, we believe that masking keywords to adaptively select task-specific lengths during inference could be a more flexible and efficient approach.
>
> > **W2**: The figure of the overall framework is not clear enough. I suggest you can bind G^T,G^V..., with the corresponding tokens in the figure.
>
> **Response:**
>
> We sincerely thank the reviewers for their detailed and thoughtful feedback, which has greatly improved the readability and completeness of our work. In the revised version, we will incorporate binding to make the definitions more precise and clearer.
>
> > **Q1&W1.2**: Lack the effectiveness of direct token selection learning without a keyword module (KM).
>
> **Response:**
>
> We conducted a complete retraining of VCM using only vision token selection configurations and then performed comparative experiments under the condition of 144 tokens. The experimental results are presented below.
>
> | Method | MME    | SciQA | RefCOCO-testA (mask) | RefCOCO-testB (mask) |
> | :----- | :----- | :---- | :------------------- | -------------------- |
> | w/ KM  | 1842.7 | 69.1  | 16.8                 | 41.6                 |
> | w/o KM | 1756.6 | 67.8  | 11.6                 | 36.8                 |
>
> As shown in the table, relying solely on vision tokens for keyword selection prevents the model from leveraging the latent space of large language models, thereby limiting its ability to search for beneficial prior information. Furthermore, this approach fails to establish a robust visual concept model, making it unsuitable for dense perception tasks beyond VQA, such as RefCOCO. Consequently, its applicability is significantly restricted.
>
> > **Q2&W1.3**: What is the average VQA score of MQT-LLaVA under the 144-token condition? Does VCM's advantage stem from differences in search space or from textual priors?
>
> **Response:**
>
> We sincerely thank the reviewers for their thorough review. As presented in Table 1 of the manuscript, we have already provided a performance comparison between MQT-LLaVA and VCM under the 144-token condition (58.6 vs. 60.8). It can be observed that VCM demonstrates a clear advantage. We believe this stems primarily from two key factors:
>
> 1) **Enhanced utilization of textual priors**: VCM benefits more effectively from the latent space of textual priors, enabling better alignment between visual features and textual features.
>
> 2) **Higher information density**: At equivalent token lengths, visual concepts carry richer information content compared to visual tokens, thereby representing more information and further boosting performance.
>
> > **Q3**:  Under 1024×1024 high-resolution images and 30 fps video streaming conditions, what are the comparative metrics of VCM versus VisionZip and SparseVLM in terms of GPU latency and memory consumption?
>
> **Response:**
>
> | Method      | Token | Flops (T) | Memory (Mb) | Avg  |
> | :---------- | :---- | :-------- | :---------- | ---- |
> | LLaVA-NeXT  | 2880  | 20.8      | 18952       | 72.4 |
> | +VisionZip  | 160   | 2.21      | 13176       | 66.2 |
> | +SparseVLM  | 160   | 2.14      | 14054       | 63.0 |
> | +VCM        | 160   | 2.07      | 13232       | 70.1 |
> | Video-LLaVA | 2048  | 14.4      | 16854       | 54.2 |
> | +VisionZip  | 136   | 1.78      | 11234       | 50.3 |
> | +SparseVLM  | 198   | 2.31      | 12876       | 46.6 |
> | +VCM        | 136   | 1.69      | 11380       | 52.5 |
>
> We sincerely thank the reviewers for their suggestions. In response, we provide a comparison of VisionZip and SparseVLM with VCM regarding FLOPs and GPU memory across high-resolution scenarios (LLaVA-Next) and video streaming (Video-LLaVA), as shown in Appendix Tables 7 and 8. From the tables, it is evident that VCM achieves the best overall results while maintaining competitive performance in both speed and memory consumption.
>
> This advantage can be attributed to the following three reasons:
>
> 1. **Model Parameter Efficiency**: VCM employs a simple two-layer attention module, which introduces minimal additional parameters compared to previous methods such as SparseVLM and VisionZip.
>
> 2. **Computational Complexity**:  VCM utilizes a forward-backward optimization algorithm, ensuring its computational complexity remains equivalent to methods[1,2,3,4] based on attention mechanisms at $O(M^2)$.
>
> 3. **Parallel Optimization**: VCM incorporates parallel optimization techniques, as detailed in Algorithm 2 of the appendix, significantly enhancing its runtime efficiency.
>
> We believe these attributes make VCM highly effective and efficient for both high-resolution and video-streaming scenarios. Thank you again for your constructive feedback, which has helped us emphasize these strengths more clearly in the revised version.
>
>
>
> [1] MQT-LLaVA: Matryoshka Query Transformer for Large Vision-Language Models
>
> [2] An Image is Worth 1/2 Tokens After Layer 2: Plug-and-Play Inference Acceleration for Large Vision-Language Models
>
> [3] PyramidDrop: Accelerating Your Large Vision-Language Models via Pyramid Visual Redundancy Reduction
>
> [4] VisionZip: Longer is Better but Not Necessary in Vision Language Models
>
>
>
> Thank you again for your insightful comments. If you have other comments, we are happy to address them to polish this work. We look forward to contributing to the development of both the multi-modal research and the open-source community.

---

> > ### Comment · Reviewer_u1dV · 2025-08-04
> >
> > Thank you for the detailed and thoughtful rebuttal. Your additional experiments and clarifications have addressed my concerns. I appreciate the thorough analysis and look forward to the updated version.

---

> > > ### Author Response · Authors · 2025-08-04
> > >
> > > We sincerely thank you for taking the time to review our paper. Your continued engagement and the valuable concerns you've highlighted are greatly appreciated. Considering the significant effort and resources our team has invested in this research, we humbly hope you might consider a score adjustment if you find our contributions deserving. Thank you again for your time and insights! If you have other comments, we are happy to address them to polish this work. We look forward to contributing to the development of both the multi-modal research and the open-source community.

---

> ### Author Response · Authors · 2025-08-04
>
> Dear Reviewer u1dV:
>
> We greatly appreciate the time and effort you dedicated to reviewing our paper. We have carefully addressed all your insightful suggestions and clarified any ambiguous points to improve our work. As the deadline for the discussion is nearing, could you kindly reconsider your evaluation based on the revised version? We are open to any further queries you might have and are eager to provide any additional information needed.
>
> Thank you for your understanding and support.
>
> Best regards,
>
> Authors

---

### Official Review · Reviewer_UGcq · 2025-07-04

**Clarity:** 3
**Significance:** 3
**Originality:** 4
**Rating:** 5
**Confidence:** 4

**Summary:**

This paper proposes a vision concept modeling (VCM) method to reduce visual tokens in VLMs. In details, VCM dynamically extracts task-relavant visual concepts (tokens) with a self-supevised instruction-finetuned technique,  and the key is a forward-backward optimization algorithm for adaptive concept granularity and spatial alignment. The proposed VCM can reduce computational costs and maintain strong performance across various vision-language tasks, including VAQ, zero-shot classification, and OV dense perception tasks.

**Questions:**

1. This paper focuses on static images/videos, it is unclear how VCM would handle temporal dynamics in complex videos?

2. Have you tested VCM on non-English datasets? How does reliance on English keyword priors affect performance in other languages?

3. Does VCM provide mechanisms to detect hallucinations or uncertain predictions when concepts are compressed (e.g., via confidence scores)?

**Ethical Concerns:**

["NO or VERY MINOR ethics concerns only"]

**Limitations:**

yes

**Quality:**

3

**Strengths And Weaknesses:**

## Strengths
- **Simple yet effective idea.** This paper introduces the first adaptive visual concept modeling method to address inefficient token-level processing in VLMs. The proposed VCM reduces computational costs (e.g., achieving up to 85% fewer FLOPs for LLaVA-1.5-7B), enabling VLMs to scale to high-resolution images.
- **Empirical validation.** Comprehensice experiments across 16 VQA datasets including image-, region-, and video-level benchmarks, zero-shot classification, OV dense prediction tasks.
- **Scalability.** The authors have validated on a series of LLaVA architectures, such as LLaVA-1.5-7/13B, LLaVA-NeXT and video-LLaVA.

## Weaknesses
- **Strong dependence on the text.** The proposed VCM relies on keywords statistics from text instructions / responses, which may introduce bias in underspecified prompts.
- Dynamic programming introduces $O(M^2)$ complexity for sequence alignment, which potentially limits scalability to long videos.

---

> ### Author Rebuttal · Authors · 2025-07-31
>
> Thanks for your professional and careful review. We respond to your concerns or questions as follows.
>
> > **W1**: Strong dependence on the text
>
> **Response:**
> The keyword selection module in VCM provides reliable keyword selection results, while the random masking strategy integrated into the VCM algorithm introduces error tolerance and robustness toward incorrect keyword predictions. Together, these two features ensure efficient and accurate learning, reducing the dependency on keyword selection. To verify these two aspects, we conducted the following experiments:
>
> To better illustrate the reliability of the keyword selection module, we randomly selected 1K samples from the 5K samples mentioned in line 85 of the paper, labeled by GPT-4o, and calculated the success rate of the module in selecting the correct keywords during training. The results are summarized below:
>
> | Training Steps | 0.5k | 1k   | 1.5k | 2k   | 2.5k | 3k   | 3.5k | 4k   | 4.5k | 5k   |
> | :------------- | ---- | :--- | :--- | :--- | ---- | ---- | ---- | ---- | ---- | ---- |
> | Accuracy       | 19.4 | 48.9 | 67.4 | 76.5 | 83.6 | 88.9 | 91.3 | 92.6 | 94.1 | 94.3 |
>
> As shown in the table, as training progresses, the keyword selection module effectively learns the relationships between text and images, ultimately achieving a 94.3% accuracy in selecting correct keywords.
>
> To further validate VCM's robustness, during inference, we randomly replaced selected keywords with non-keywords at varying replacement probabilities *r* to simulate unreliable predictions by the keyword selection module. We then tested the model and calculated the average visual concept token length across SciQA, POPE, and MME datasets. The results are summarized below:
>
> | r    | Mean Token | SciQA | POPE | MME    |
> | :--- | :--------- | :---- | ---- | ------ |
> | 0%   | 64         | 69.1  | 85.2 | 1719.6 |
> | 15%  | 76         | 68.9  | 85.1 | 1720.3 |
> | 30%  | 89         | 69.0  | 84.9 | 1725.4 |
> | 45%  | 96         | 68.7  | 85.0 | 1726.8 |
> | 60%  | 103        | 68.9  | 85.1 | 1728.7 |
>
> From the table, it can be observed that VCM’s masking strategy ensures strong error tolerance. When the keyword selection module produces incorrect predictions, VCM tends to generate longer visual concepts to ensure the downstream LLM receives sufficient prior information for task reasoning.
>
> > **W2&Q1**: Dynamic programming introduces O(M2) complexity for sequence alignment, which potentially limits scalability to long videos. This paper focuses on static images/videos, it is unclear how VCM would handle temporal dynamics in complex videos?
>
> **Response:**
>
> Compared to previous methods based on attention mechanisms[1,2,3,4], VCM has the same theoretical computational complexity of *O*(*M*^2), with no significant difference in this regard. However, VCM can be effectively combined with downsampling techniques, where long videos are first downsampled and then further compressed using VCM. This combination allows VCM to be efficiently applied to long video understanding tasks. To further validate the scalability of our method in complex long video scenarios, we selected the challenging long-video benchmark dataset, Video-MME, for evaluation and comparison. Specifically, we processed 64 frames per input, and the experimental results are illustrated in the figure below.
>
> | Method      | Token$\cdot$Frames | Video-MME |
> | :---------- | :----------------- | :-------- |
> | Video-LLaVA | 256$\cdot$64       | 42.1      |
> | VCM         | 17$\cdot$64        | 40.9      |
>
> Compared to the baseline method Video-LLaVA, VCM reduces the input sequence length for a single frame from 256 to 17, while maintaining strong performance on the Video-MME benchmark. This demonstrates that our method, like previous approaches, is plug-and-play and exhibits excellent scalability.
>
> > **Q2**: Have you tested VCM on non-English datasets? How does reliance on English keyword priors affect performance in other languages?
>
> **Response:**
>
> To investigate the impact of non-English data on VCM's generalization capability, we collected the LLaVA-ZH Chinese dataset. We mixed it with the original LLaVA dataset and trained the model for 500 steps on the combined dataset. The model's performance was then tested on MMB-ZH and MMB-EN, and the experimental results are presented below.
>
> | Method         | Token | MMB-zh | MMB-en |
> | :------------- | ----- | :----- | :----- |
> | LLaVA+llava-zh | 576   | 44.6   | 46.5   |
> | VCM+llava-zh   | 144   | 43.3   | 45.8   |
>
> As shown in the table, VCM can successfully train and test on Chinese data, achieving performance comparable to that on English data. This demonstrates VCM's strong cross-linguistic generalization capability.
>
> > **Q3**:Does VCM provide mechanisms to detect hallucinations or uncertain predictions when concepts are compressed (e.g., via confidence scores)?
>
> **Response:**
>
> We conducted an ablation study to evaluate the impact of different confidence score thresholds on model performance during conditional reasoning after training for 500 steps, where the thresholds were also used to detect and filter uncertain information, influencing the model's performance.
>
> | Thres | SciQA | VizWiz | POPE | MME    | RefCOCO-testB (mask) |
> | :---- | :---- | :----- | :--- | ------ | -------------------- |
> | 0.5   | 64.1  | 48.2   | 76.6 | 1350.6 | 26.8                 |
> | 0.55  | 64.6  | 49.8   | 76.8 | 1355.2 | 27.3                 |
> | 0.6   | 64.7  | 49.7   | 76.8 | 1354.7 | 27.4                 |
> | 0.65  | 64.4  | 49.5   | 76.4 | 1353.3 | 27.0                 |
>
> As shown in the table, increasing the confidence score threshold during inference helps filter out noisy and uncertain information, thereby improving the model's performance on downstream tasks. This indicates that the confidence scores in our end-to-end optimization design carry practical physical significance, aligning with the design principles stated in line 176 of the paper. Specifically, confidence scores represent the uncertainty in concept compression and can be effectively utilized to design mechanisms for detecting hallucinations or uncertain predictions.
>
> [1] MQT-LLaVA: Matryoshka Query Transformer for Large Vision-Language Models
>
> [2] An Image is Worth 1/2 Tokens After Layer 2: Plug-and-Play Inference Acceleration for Large Vision-Language Models
>
> [3] PyramidDrop: Accelerating Your Large Vision-Language Models via Pyramid Visual Redundancy Reduction
>
> [4] VisionZip: Longer is Better but Not Necessary in Vision Language Models
>
>
>
> Thank you again for your insightful comments. If you have other comments, we are happy to address them to polish this work. We look forward to contributing to the development of both the multi-modal research and the open-source community.

---

> > ### Author Response · Authors · 2025-08-04
> >
> > Dear Reviewer UGcq:
> >
> > We are truly grateful for your insightful comments and the guidance you provided during your review of our paper. We are pleased to inform you that we have addressed all points raised and have made significant improvements. As the discussion phase draws near, we kindly request your reevaluation at your earliest convenience. Should any questions remain, we are at your disposal to clarify them promptly.
> >
> > Thank you for your time and understanding.
> >
> > Sincerely,
> >
> > Authors

---

> ### Author Response · Authors · 2025-08-07
>
> Dear Reviewer UGcq:
>
> Thank you again for your time and valuable feedback on our paper. We noticed that your official comment on OpenReview is still pending. Could you please let us know if there’s any additional information needed from our side? We’d greatly appreciate it if you could submit your comment when you have a moment.
>
> Sincerely,
>
> Authors

---

> ### Author Response · Authors · 2025-08-08
>
> Dear Reviewer UGcq,
>
> I hope this finds you well. As we near the end of the discussion period, we wanted to thank you once again for your thoughtful review. Please let us know if there are additional questions or aspects of the work you’d like us to address.
>
> We highly value your expertise and feedback, and we’re eager to continue discussing any unresolved points that could help refine and improve our research.
>
> Thank you for your time and for contributing to the review process.
>
> Sincerely,
>
> Author

---

### Official Review · Reviewer_DmHA · 2025-07-07

**Clarity:** 2
**Significance:** 3
**Originality:** 4
**Rating:** 5
**Confidence:** 5

**Summary:**

The paper introduces Visual Concept Model (VCM), a new type of large vision language models (LVLMs) where visual embeddings focus on semantic concepts, forming a shorter sequence. The training procedure of VCM follows the standard pre-training and instruction tuning pipeline. In the pre-training stage, the authors propose to train an additional keyword selection module via a semantic alignment loss, and a multi-head cross-attention layer to enhance visual representation. During instruction tuning, the keyword selection module identifies keywords in the text, which are then used to compute the proposed VCM loss. Through experiments, the authors reveal a negative correlation between the required visual token length and the keyword number difference between text instructions and responses. This relationship helps estimate the sequence length of the visual concept embeddings. Finally, vision concept embeddings are selected based on a learned assignment probability, optimized via the VCM loss with the help of dynamic programming. Extensive experiments across diverse benchmarks and tasks are conducted to validate VCM’s effectiveness.

**Questions:**

The reviewer is uncertain about the following details:
1. What is the objective of the binary classification in equation (3)? How to train the linear $f_{CLS}$?
2. How to obtain each probability $p(z_l|y_t)$ when conducting forward and backward probability transition in section 3.3?
3. Please explain the implicit contrastive sampling module and the corresponding training process. These content seem missing in the main text and only appear in the caption for Figure 3.
4. How does the proposed method address the first key challenge mentioned in line 73?

**Ethical Concerns:**

["NO or VERY MINOR ethics concerns only"]

**Final Justification:**

The authors provide detailed explanation to the ambiguities in the original text, which had addressed my concerns. I am willing to maintain my recommendation for acceptance because of the novelty of the method, and raise my score for the better clarity. The authors should refine the manuscript as promised after the rebuttal.

**Limitations:**

yes

**Paper Formatting Concerns:**

Not formatting issues.

**Quality:**

3

**Strengths And Weaknesses:**

Strengths:
1. The proposed method is well-motivated and novel. Representing the visual inputs by a shorter yet more semantically meaningful visual embedding sequence can address the sequence length limitation of current LVLMs and narrow their gap to human cognition. The paper introduces several novel modules and optimization objectives to obtain the visual concept embeddings that potentially shed lights to developing new LVLMs.
2. The analytical experiment that reveals the correlation between keyword number and minimum visual sequence length is interesting.
3. The evaluation is comprehensive with promising results.

Weaknesses:
While the proposed method is novel, its presentation could be enhanced by simplifying notations and incorporating additional details for clarity. Current presentation increases the difficulty for fully understand the paper. The reviewer may consider lower the score if the following concerns are not well addressed in the rebuttal.
1. The notations are complicated:
 - The keyword selection module $f_{KS}$ and  'MHSA' in line 102 and equation (2) appears to be the same thing. Consider unifying these notations.
 - Similarly, consider simplifying $f_{V2T}$ in section 3 to $f_M$.
 - The text prior $T$ in line 121 is undefined, which the reviewer assumes to be the keywords.
 - Typo in line 121.
 - Consider using a notation like $f_{T}$ to represent the LLM's word embeddings (by regarding them as an embedding function) instead of $X^T$ to align with other notations.
2. Several components are left unexplained. Please refer to the questions.

---

> ### Author Rebuttal · Authors · 2025-07-31
>
> Thanks for your professional and careful review. We respond to your concerns or questions as follows.
>
> > **W1**: The notations are complicated.
>
> **Response:**
>
> Your careful attention to the consistency and clarity of the notation has been invaluable in helping us improve the quality and readability of our work. We fully agree with all your suggestions and are grateful that you took the time to identify these important improvements.
>
> In the revised version, we will: (1) unify $f_{KS}$ and 'MHSA' in line 102 and Equation (2) to use $f_{KS}$ consistently; (2) simplify $f_{V2T}$ to $f_M$ in Section 3; (3) add a clear definition for the text prior $T$ in line 121 (confirming it refers to keywords); (4) fix the typo in line 121; and (5) use $f_T$ instead of $X^{\top}$ to represent the LLM's word embeddings for better notational consistency.
>
> We are committed to implementing all these changes carefully. Thank you for your constructive feedback.
>
> > **Q1**: What is the objective of the binary classification in equation (3)? How to train the linear $f_{CLS}$?
>
> **Response:**
>
> Thank you for your comments. In Section 3.3, we have provided a detailed explanation of the training process. Here, we would like to further elaborate on the specific steps involved.
>
> The binary classification in Equation (3) generates binary logits $Y^V$, which indicate whether each visual token should be retained. These logits are combined with the target sequence $Z^V$—obtained by expanding the token length using the prior distribution patterns illustrated in Figure 2—and subjected to optimization through the forward-backward algorithm. During this end-to-end training process, the linear classifier $f_{CLS}$ is trained to effectively learn the mapping between visual tokens and their likelihood of being selected.
>
> > **Q2**: How to obtain each probability $p(z_l|y_t)$ when conducting forward and backward probability transition in section 3.3?
>
> **Response:**
>
> Thank you for the feedback. Here, we provide further clarification.
>
> Firstly, the logits $Y^V = \[ y_t^V \]^M_{t=1}$ are produced as outputs from the linear classifier $f_{CLS}$. Then, the probabilities $p(z_l^V | y_t^V)$ are computed using $p(z_l^V | y_t^V) = \text{Softmax}(Y^V)$. Essentially, this represents a binary classification probability since $z_l^V \in \[\star, \emptyset\]$.
>
> We will incorporate this explanation into the paper to make the presentation clearer and more accessible. Thank you once again for your insightful comments, which have been instrumental in enhancing the clarity of our work.
>
> > **Q3**: Please explain the implicit contrastive sampling module and the corresponding training process. These content seem missing in the main text and only appear in the caption for Figure 3.
>
> **Response:**
>
> As discussed in lines 125–130 of the main text, this module fundamentally serves as a data augmentation mechanism and does not contain trainable parameters. Specifically, we randomly apply masking to the extracted keywords, and VCM leverages the masked data to generate extended visual concepts. This ensures that the subsequent LLM processing does not encounter hallucinations or conflicts due to missing information.
>
> In Table 4, we conducted ablation experiments to validate this masking strategy. The results demonstrate that this approach introduces greater data diversity without adding additional risks, significantly enhancing the effectiveness of VCM’s training and improving its final performance.
>
> Additionally, we will include references to  Figure 3 in the revised version to ensure smoother readability and make the explanation more intuitive for readers. Thanks for your valuable insights, which have helped us refine and improve the clarity of our work.
>
> > **Q4**:How does the proposed method address the first key challenge mentioned in line 73?
>
> **Response:**
>
> We sincerely appreciate the reviewers' valuable feedback. Here, we provide further clarification on this point.
>
> By utilizing VCM's forward-backward optimization algorithm, we can efficiently perform self-supervised learning on large-scale VQA datasets that lack fine-grained annotations. This process allows us to construct a robust visual concept model. Furthermore, we employ the implicit contrastive sampling module as a data augmentation technique, which further increases the diversity of the training data and enhances the effectiveness of visual concept modeling.
>
> With these two contributions, we address the first key challenge mentioned in line 73. Unlike previous methods, which require fine-grained annotations—not only identifying the keywords in the text but also aligning each keyword with specific regions in the image-our approach eliminates the dependency on such detailed labels, making it both more scalable and efficient.
>
>
>
> Thank you again for your insightful comments. If you have other comments, we are happy to address them to polish this work. We look forward to contributing to the development of both the multi-modal research and the open-source community.

---

> > ### Author Response · Authors · 2025-08-04
> >
> > Dear Reviewer DmHA:
> >
> > We hope this message finds you well. We deeply appreciate your thoughtful feedback and the attention you’ve given to our manuscript. All concerns have been thoroughly addressed, and we wish to invite you to review the manuscript once more. With the deadline approaching, we would be grateful if you could confirm that all uncertainties have been resolved. We are ready to assist you with any further clarifications.
> >
> > Thanks for your cooperation.
> >
> > Kind regards,
> >
> > Authors

---

> > ### Comment · Reviewer_DmHA · 2025-08-06
> >
> > Thank you for the detailed explanation, which had helped me better understand the paper. The rebuttal successfully addresses my concerns. Considered the novelty of the paper and its potential to appeal to a broad readership, I maintain my recommendation for acceptance and raise my score to 5.

---

### Author Response · Authors · 2025-08-01

We appreciate the reviewers’ insightful comments and constructive feedback on our manuscript. We are pleased to receive positive ratings from most of the reviewers. Furthermore, we are delighted to learn that the reviewers found the research problem to be significant and the core idea to be interesting (Reviewers DmHA, UGcq, RMxa, and u1dV), the technical methodology to be novel and sound (Reviewers DmHA, UGcq, and RMxa), and the experiments to be convincing and comprehensive (Reviewers UGcq, RMxa, and u1dV). Based on the reviews, we provide a general response to the points raised by multiple reviewers and individual responses below to address each reviewer’s concerns.

**(1)** Regarding the questions about the experiments, we have taken the following actions:

*   For Reviewers DmHA, UGcq, u1dV, and RMxa, we have either emphasized the location of the required experiments for corresponding comments in our paper or added experiments correspondingly.

*   For Reviewer RMxa, we clarified the seemingly counter-intuitive results in Table 4 (e.g., fewer tokens outperforming more tokens in some benchmarks) and explained the domain-specific task requirements for token lengths. Further ablations confirmed the robustness of our token merging approach.

*   For Reviewer UGcq, we evaluated VCM on non-English datasets, providing results on the Chinese dataset LLaVA-ZH. The experiments demonstrate VCM’s strong cross-linguistic generalization, comparable to English benchmarks.

*   For Reviewers UGcq and u1dV, we analyzed VCM’s scalability to long video tasks and high-resolution images (e.g., tested on Video-MME and 1024×1024 benchmarks). The results verify VCM’s competitive memory efficiency and FLOPs reduction across such scenarios.

*   For Reviewer u1dV, we included experiments comparing keyword-supported vision concept modeling against methods omitting the keyword module. Results validate the necessity and impact of using text priors for aligning visual concepts.

**(2)** We have addressed the questions about the idea and technical details as follows:

*   For Reviewer DmHA, we clarified the binary classification objective in Equation (3) and the forward-backward probability transition process, elaborating on their role in vision concept selection.

*   For Reviewers DmHA and RMxa, we further discussed the role of the implicit contrastive sampling module and explained its augmentation strategy in training VCM for error tolerance and robustness.

*   For Reviewers UGcq and RMxa, we analyzed and quantified the computational overhead of VCM’s key components (e.g., keyword selection module and dynamic programming) compared to the baseline LLMs. We demonstrated that VCM introduces minimal latency while maintaining significant computational savings.

*   For Reviewer RMxa, we explained why freezing the vision encoder yields stable performance by preserving pre-trained knowledge, aligning this with standard practices from existing LLaVA-based approaches.

**(3)** Missing or extended analysis:

*   For Reviewer u1dV, we expanded the analysis of the threshold $ \tau $ used in length estimation and conducted sensitivity analysis to validate the robustness of VCM’s heuristic decisions.

*   For Reviewer RMxa, we justified our choice of heuristic-based length estimation, showing minimal performance differences compared to ground-truth lengths. We attributed this to VCM’s error tolerance and forwarding-backward optimization.

We sincerely thank all the reviewers for their constructive suggestions. Please feel free to let us know if further details/explanations would be helpful.

Yours truly,
Authors of #1035

---

### Comment · Area_Chair_9Ze9 · 2025-08-05

As the Author-Reviewer discussions will end on Aug 8, please read the author's response and engage in the discussion with the authors.

- Are the authors' answers satisfactory for your questions?
- If so, would you modify your score accordingly?
- If not, could you leave the reasons?

AC.

---

### Note · Authors · 2025-08-11

Dear Program Chairs (PC), Senior Area Chairs (SAC), Area Chairs (AC), and Reviewers,

We sincerely thank all the reviewers for their constructive feedback, which has greatly refined our manuscript. We are encouraged by the overall positive evaluations and thoughtful comments from the reviewers, and are deeply grateful for the valuable suggestions. These suggestions have helped us clarify and improve key aspects of our work. This broad recognition further highlights the significance and robustness of our approach.

The core contribution of our work lies in introducing a paradigm shift in the field of vision token compression. Our method not only achieves significant performance improvements over previous state-of-the-art (SOTA) approaches with a more efficient compression strategy, but also redefines the problem of vision token compression as one of fundamental visual concept extraction. By leveraging the VCM algorithm, which can be trained in a self-supervised manner, our approach significantly enhances the model's ability to perceive visual concepts, greatly expanding its potential applications. We further clarify that our method can be directly applied to large-scale unannotated vision-language instruction datasets, enabling efficient visual concept learning through masked augmentation strategies and the self-supervised VCM algorithm. Moreover, VCM demonstrates strong generalization and transferability across visual encoders of various sizes and multiple dense perception vision-language downstream tasks, highlighting its wide applicability.

Finally, we once again sincerely thank all the reviewers for their invaluable suggestions, which have greatly improved the scientific rigor and clarity of our manuscript. If the paper is accepted, we will carefully incorporate all feedback into the final version. We believe our work provides a novel and effective paradigm for the field of vision token compression and opens up new possibilities for the design of dynamic tokenizers.

Sincerely,
The Authors

---

### Decision · Program_Chairs · 2025-09-17

**Decision:**

Accept (poster)

**Comment:**

This paper proposes a novel method, namely "vision concept modeling," to address token-level redundancy for high-resolution images in Vision-Language Models. By learning visual concepts in a self-supervised manner, the proposed method significantly reduces computational costs while maintaining strong performance on downstream tasks. Reviewers acknowledged this as a valuable and timely contribution to the field.

While the initial reviews raised constructive questions regarding ablation studies, experimental comparisons, and methodological heuristics, the authors provided a thorough and effective rebuttal that addressed these points comprehensively. The reviewers were satisfied with the authors' responses and engaged productively during the discussion period. There is a clear consensus that this will be a solid paper once the promised revisions are incorporated into the final manuscript. Therefore, the recommendation is Accept.